# PRDP: Progressively Refined Differentiable Physics

**Kanishk Bhatia**[*], **Felix Koehler**[*1], **Nils Thuerey**
Technical University of Munich
{kanishk.bhatia,f.koehler,nils.thuerey}@tum.de
[*]Equal contribution

## ABSTRACT

The physics solvers employed for neural network training are primarily iterative, and hence, differentiating through them introduces a severe computational burden as iterations grow large. Inspired by works in bilevel optimization, we show that full accuracy of the network is achievable through physics significantly coarser than fully converged solvers. We propose *progressively refined differentiable physics* (PRDP), an approach that identifies the level of physics refinement sufficient for full training accuracy. By beginning with coarse physics, adaptively refining it during training, and stopping refinement at the level adequate for training, it enables significant compute savings without sacrificing network accuracy. Our focus is on differentiating iterative linear solvers for sparsely discretized differential operators, which are fundamental to scientific computing. PRDP is applicable to both unrolled and implicit differentiation. We validate its performance on a variety of learning scenarios involving differentiable physics solvers such as inverse problems, autoregressive neural emulators, and correction-based neural-hybrid solvers. In the challenging example of emulating the Navier-Stokes equations, we reduce training time by 62%.

## 1 INTRODUCTION

*Differentiable Physics* is a paradigm which allows learning algorithms to interact with gradients of classical physics solvers. This has proven effective across many domains, e.g., solving inverse problems (Bendsoe & Sigmund, 2013), integrating physical constraints (Raissi et al., 2019; Li et al., 2024), and especially, creating hybrid models that blend classical numerical techniques with learned components (Um et al., 2020; Kochkov et al., 2021; 2024). Despite their promise, neural-hybrid models for differential equations face limited adoption due to the computational cost of executing and differentiating through classical solvers during training. At the core of most classical solvers for differential equations are iterative processes that can be tuned for accuracy, typically by adjusting parameters such as step size or iteration count. Traditionally, these methods prioritize achieving the highest possible accuracy in the physics solver. In contrast, our work takes a novel approach, drawing inspiration from bi-level optimization (Pedregosa, 2016). Rather than focusing solely on maximum physics accuracy, we explore how numerical solvers can be strategically adjusted to substantially accelerate the training process without losing network accuracy.

Our work applies to training pipelines involving differentiable physics, as exemplified in figure 1. Repeatedly querying a solver with several iterations in the forward pass – and differentiating through its iterations in the backward pass – introduces a severe computational bottleneck during training. Since deep learning is inherently based on noisy gradient estimates, we show that neither the physics nor the physics' Jacobian must be fully converged at training

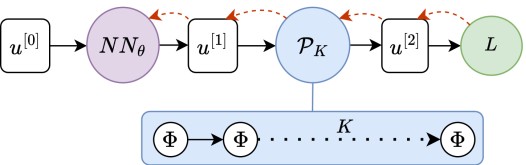

Figure 1: A neural network training pipeline using a differentiable physics solver $\mathcal{P}_K$. Black and red arrows show the forward and backward passes, respectively. As solver iterations $K$ grows, the cost of passes through $\mathcal{P}_K$ becomes severe.

---

[1]Munich Center for Machine Learning

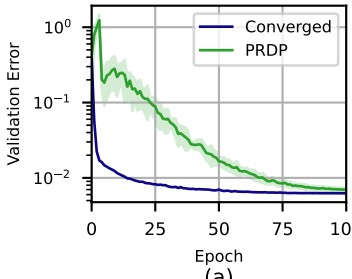 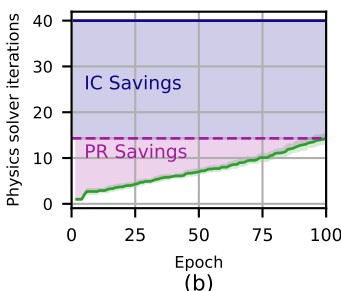 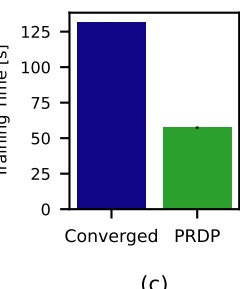

Figure 2: *Progressively Refined Differentiable Physics* (PRDP) reduces the training time of neural networks containing numerical solver components (c). The fidelity of iterative components is increased only if validation metrics plateau. This leads to savings by using fewer iterations in the beginning (PR savings in (b)) and by ending at a refinement level significantly below full fidelity (IC savings in (b)). The achieved validation error is identical (a).

time to attain a good generalization. Using differentiable numerical solvers at a level significantly coarser than needed for tight tolerances and progressively refining it starting from an even coarser level is sufficient to achieve full accuracy of the network.

Differentiable Physics can be understood as a type of bilevel optimization. In the context of hyper-parameter optimization (a typical bilevel optimization problem), Pedregosa (2016) explored the idea of successively refining an inner solver to make increasingly accurate updates in an outer optimizer. Under strict assumptions of convexity, it was shown that with a summable sequence of refinement levels (in terms of increasingly tighter tolerances for both forward and backward solve), convergence in the low-dimensional hyperparameter space can be achieved. We build upon this work but instead, treat the nonconvex learning of neural network parameters as an outer problem. Our inner problem is the most elementary operation in any scientific computing, the solution of a linear system of equations from the discretization of a physical model. In this more general setting, we observe that we cumulatively reduce inner iterations not only by progressive refinement but also by ending the refinement at a level significantly coarser than needed for full physics convergence. This is possible because of the approximative nature of neural network training for which highly accurate gradients are not required. Similar approaches have been investigated for Deep Equilibrium Models (Bai et al., 2019), which have nonlinear root-finding with dense Jacobians as the inner problem, e.g., by (Shaban et al., 2019; Fung et al., 2021; Geng et al., 2021). In contrast, we focus on the sparse linear systems arising from the discretization of partial differential equations (PDEs) admitting special solution characteristics that have been unexplored in the context of deep learning. For realistically large and sparse linear systems of equations, the prevalent class of solution methods are iterative linear solvers (Saad, 2003). By controlling the number of iterations in these solvers (and during their backward passes), we can directly balance physics refinement and computational cost.

Reducing the cumulative number of solver iterations performed throughout the network training results in considerable compute savings since linear solves typically dominate run times. However, the optimal refinement schedule and the sufficient level of refinement are problem-dependent. To automatically determine them during the network training, we present a novel algorithm, *Progressively Refined Differentiable Physics*, in which the refinement of the physics is adaptively increased if a plateau in terms of validation metrics is encountered. This idea is visualized in figure 2.

Our experiments focus on efficiently training neural networks with differentiable linear solvers in the loop. We address *unrolled* as well as *implicit* differentiation methods, showing that PRDP applies effectively to both. The approach is tested on training tasks across a range of PDE problems, including the Poisson, heat diffusion, Burgers, and Navier-Stokes equations. Empirical insights into PRDP's behavior are presented for 1D problems, and its performance is validated through more complex 2D and 3D time-stepping problems. We demonstrate its effectiveness in a variety of settings such as inverse problems, differentiable physics losses, and correction-based approaches.

In summary, our main contributions are the following.

- We empirically demonstrate that full network performance can be achieved with a coarse level of physics refinement, well below the typical refinement required for full convergence, leading to significant computational savings.

- We introduce the *Progressively Refined Differentiable Physics (PRDP)* algorithm, which adaptively identifies the optimal level of physics refinement during training.
- We validate the effectiveness of PRDP across various differentiable physics learning scenarios, demonstrating its broad applicability.

## 2 DIFFERENTIATING ITERATIVE LINEAR SOLVERS

A theoretical understanding of differentiating iterations can be gained through the example of an inverse problem involving a parameterized linear system $\boldsymbol{A}\boldsymbol{u}_h = \boldsymbol{b}_h$. Such systems arise in the numerical solution of discretized PDEs, where $h$ refers to the spatial discretization width (Ames et al., 2014). For this, both the system matrix $\boldsymbol{A}$ and the right-hand side $\boldsymbol{b}_h$ arise from the assembly routines $\Lambda(\theta)$ and $\beta(\theta)$, respectively. An iterative solver $\Phi$ creates a sequence of guesses $\{\boldsymbol{u}_h^{[0]}, \boldsymbol{u}_h^{[1]}, \dots \boldsymbol{u}_h^{[K]}\} = \{\Phi^k(\boldsymbol{u}_h^{[0]}; \boldsymbol{A}, \boldsymbol{b}_h)\}_{k=0}^{K}$ that should converge to the analytical solution $\boldsymbol{u}_h^{[K]} \approx \boldsymbol{u}_h^* = \boldsymbol{A}^{-1}\boldsymbol{b}_h$, up to a given tolerance $\epsilon$. We denote the number of solver iterations required to achieve this tolerance by $K_\epsilon$. Clearly, the set of parameters $\theta \in \mathbb{R}^P$ affect the solution to this linear system of equations. Let $\mathcal{P}_K$ denote a function mapping from the parameters $\theta$ to the iterative solution of the linear system of equations $\boldsymbol{u}_h^{[K]}$ via first assembling the system matrix $\boldsymbol{A} = \Lambda(\theta)$ and the right-hand side $\boldsymbol{b}_h = \beta(\theta)$ and then employing the iterator $\Phi$ for $K$ steps. An example of the corresponding compute graph is shown in figure 21 in the appendix. The inverse problem is solved by performing an optimization over the parameter space, aiming to minimize the discrepancy against a reference solution $\boldsymbol{u}_h^r$ via

$$\min_\theta \quad L(\theta) = l(\boldsymbol{u}_h(\theta)) = \frac{1}{2}\|\boldsymbol{u}_h(\theta) - \boldsymbol{u}_h^r\|_2^2 \quad \text{s.t.} \quad \Lambda(\theta)\boldsymbol{u}_h = \beta(\theta). \tag{1}$$

This forms a bilevel problem where the outer optimization concerns minimizing the loss $L(\theta)$, while the inner optimization pertains to solving the linear system $\mathcal{P}_K$. Then, the loss function's evaluation can be explicitly written as $L(\theta) = l(\mathcal{P}_K(\theta))$.

Solving the outer optimization using a first-order method requires the gradient $\nabla_\theta L$. We can differentiate this chained function using reverse-mode automatic differentiation (AD, Griewank & Walther (2008)) to find the (transposed) gradient as

$$(\nabla_\theta L)^T = \bar{\theta}^T = \left( \vec{1}^T \underbrace{\boldsymbol{J}_l|_{\mathcal{P}_K(\theta)}}_{\text{primal inacc.}} \right) \underbrace{\boldsymbol{J}_{\mathcal{P}_K}|_\theta}_{\text{adjoint inacc.}}. \tag{2}$$

Clearly, the quality of the gradient depends on the number of iterator steps $K$. Inaccuracies in the physics operator $\mathcal{P}_K$ propagate to the loss gradient $\nabla_\theta L$ through two sources: primal inaccuracy, i.e., the Jacobian of the loss function $\boldsymbol{J}_l$ evaluated at the approximate solution $\mathcal{P}_K(\theta)$, and adjoint inaccuracy, i.e., the Jacobian $\boldsymbol{J}_{\mathcal{P}_K}|_\theta$ of the (approximate) iterative solver itself.

### 2.1 LINEAR SOLVER VJPS

AD frameworks do not assemble the full Jacobian; rather, they employ vector-Jacobian products (VJP) to reverse-propagate the gradient information (Murphy, 2023; Blondel & Roulet, 2024). The programmatic implementation for the VJP of the loss function $\bar{\boldsymbol{u}}_h^T = \vec{1}^T \boldsymbol{J}_l|_{\mathcal{P}_K(\theta)}$ is straightforward to perform via AD. On the other hand, the VJP over the approximate physics $\mathcal{P}_K$ requires reverse propagation over the solver and the assembly routines, which we detail in appendix C. Conceptually, there are two approaches.

**Implicit Differentiation** The backpropagation over the iterative linear solve can be framed as the solution of another linear system in terms of an auxiliary variable $\lambda$ with $\boldsymbol{A}^T\lambda = \bar{\boldsymbol{u}}_h$. This requires the transpose system matrix $\boldsymbol{A}^T$. Implicit differentiation over any kind of implicit relation gives rise to a linear solve with the system's linearized form being transposed. It can be derived automatically with AD tools (Blondel et al., 2022). Having a linear solve in the primal execution as well is a special case, as the same system matrix appears in the forward as well as the backward pass. Hence, it is reasonable to employ the same iterator in the VJP as in the primal, but with the transposed system matrix and the different right-hand side $\Phi(\cdot; \boldsymbol{A}^T, \bar{\boldsymbol{u}}_h)$. The number of iterates required to converge can be different from the primal and provide a way to control the adjoint inaccuracy.

**Unrolled Differentiation** Unrolled differentiation applies AD directly to an iterative program, treating each iteration as an individual computational step. It thereby accesses the VJP through the iterator $\Phi$ and accumulates each iterate's contribution. This inherently requires access to all primal iterates. Since the AD engine unrolls as many iterations as in the primal pass, the adjoint accuracy is naturally coupled with the primal accuracy.

## 2.2 SCHEDULING INNER ITERATIONS

Since the outer problem of equation 1 is solved iteratively, the gradient $\nabla_\theta L$ can be coarse at the beginning of the outer iterations and only requires the highest fidelity towards the end (Pedregosa, 2016). This fundamental idea was developed in the domain of hyperparameter optimization, and we demonstrate its application to physical inverse problems. Consider the Poisson equation, which is a prototypical elliptic partial differential equation found in many areas of science and engineering. Most discretization techniques lead to a linear system of equations with $N$ degrees of freedom (for an example discretization, refer to section D.1).

$$\Delta u(x) = -p(x, \theta), \ x \in \Omega \subset \mathbb{R}^D \quad \implies \quad \boldsymbol{A}\boldsymbol{u}_h = \boldsymbol{b}_h, \ \boldsymbol{u}_h, \boldsymbol{b}_h \in \mathbb{R}^N, \boldsymbol{A} \in \mathbb{R}^{N \times N} \quad (3)$$

We consider a setting on the one-dimensional unit interval $\Omega = (0, 1)$ with homogeneous Dirichlet boundary conditions, $u(0) = 0 = u(1)$. Assuming the parameter space is one-dimensional, we design the right-hand side as $p(x, \theta) = \theta \sin(2\pi x)$ and discretize it on the domain. The outer optimization problem over $\theta \in \mathbb{R}$ can be exactly solved in this case.

We employ a Jacobi scheme to solve the linear system which converges to $\epsilon = 10^{-5}$ within $K_\epsilon = 600$ iterations; so does its unrolled Jacobian. Under a gradient descent optimizer in the parameter space of $\theta$, it requires a total of 125 outer iterations to bring the relative parameter suboptimality against $\theta^r$ also down to $10^{-5}$. By using simple scheduling that starts the outer optimization at a coarse inner resolution of $K = 25$, and progressively increases the inner iterations by $\Delta K = 6$ in every outer iteration, all the way to $K_\epsilon$, we can reduce the overall computational cost of the optimization while achieving the same parameter suboptimality. These results are visualized in figure 3. Despite requiring slightly more outer iterations, fewer inner iterations are necessary. This results in an overall reduction of 28% of the total number of inner iterations. These savings represent one component of the final cost savings. As they are obtained due to scheduling fidelity of the physics solve, we denote them as *progressive refinement* (PR) savings.

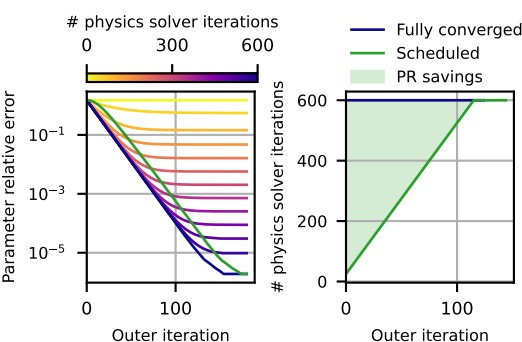

Figure 3: Progressively refining the differentiable physics during outer optimization of a Poisson inverse problem achieves full convergence of the parameter with fewer cumulative iterations of the physics solver, leading to *progressive refinement* (PR) savings.

## 2.3 NETWORK TRAINING UNDER INCOMPLETELY CONVERGED DIFFERENTIABLE PHYSICS

Going beyond simple convex inverse problems, more sophisticated compute graphs arise. For example, assume the linear system is assembled from a prior variable that is given as the output of a neural network $\boldsymbol{g}_h := \boldsymbol{f}(\cdot; \theta)$ (ignoring the input to the network for now). In this case, we have $\boldsymbol{A} = \Lambda(\boldsymbol{g}_h)$ and $\boldsymbol{b}_h = \beta(\boldsymbol{g}_h)$. Hence, the optimization over $\theta$ turns into the nonconvex learning problem in the weight space of the neural network. The extended reverse-mode AD operation reads

$$(\nabla_\theta L)^T = \bar{\theta}^T = \left( \left( \vec{\boldsymbol{1}}^T \ \boldsymbol{J}_l|_{\mathcal{P}_K(\boldsymbol{f}(\cdot; \theta))} \right) \ \boldsymbol{J}_{\mathcal{P}_K}|_{\boldsymbol{f}(\cdot; \theta)} \right) \boldsymbol{J}_{\boldsymbol{f}}|_\theta. \quad (4)$$

The approximate solution to the linear system of equations using $K$ steps and its differentiation remain the sources of gradient inaccuracy. Yet, the effect of the neural network alters the *solution characteristics* of the iterative linear solver through the assembly of the system matrix (influencing its spectrum) and right-hand side.

Conversely, we hypothesize that neural network training can also work under approximate gradients. To illustrate this, consider the one-dimensional heat equation on a periodic unit interval with a time-implicit discretization

$$\frac{\partial u}{\partial t} = \nu \frac{\partial^2 u}{\partial x^2}, \; x \in (0, 1) \implies (\boldsymbol{I} - \nu \Delta t \, \mathcal{L}_1) \boldsymbol{u}_h^{[t+1]} = \boldsymbol{u}_h^{[t]}, \tag{5}$$

where $\mathcal{L}_1$ represents the matrix from the spatial discretization of the second derivative (see section D.2 for more details). The physics operator now advances from one step to the next $\boldsymbol{u}_h^{[t+1]} = \mathcal{P}(\boldsymbol{u}_h^{[t]})$.[2] Consider the scenario in which a neural network performs a first prediction in time from an initial state, i.e., $\boldsymbol{u}_h^{[1]} = \boldsymbol{f}(\boldsymbol{u}_h^{[0]}; \theta)$, followed by the physics operator involving $K$ iterator steps which provides the second time step solution $\boldsymbol{u}_h^{[2][K]} = \mathcal{P}_K(\boldsymbol{u}_h^{[1]})$. The loss in equation 1 is computed against a reference given by applying the converged physics operator twice $\boldsymbol{u}_h^{[2],r} = \mathcal{P}_{K_\epsilon}^2(\boldsymbol{u}_h^{[0]})$. Hence, the network is trained to emulate one application of $\mathcal{P}$. We solve the linear system of equations using the Jacobi method (B.1), which converges within $K_\epsilon = 25$ iterations. The implicit differentiation's linear solve requires equally many iterations.

Figure 4 shows the results of multiple network training runs performed at different levels of physics refinement, i.e., different values of $K$ that are kept constant for each training run. We observe that beyond a refinement threshold $K_{\max}$, no further improvements of the neural emulator's performance on a validation metric are noticeable. Hence, we conclude that networks can attain a high level of accuracy, even when trained through *incompletely converged* (IC) physics. As explained in equation 2, this incomplete physics convergence subsumes coarseness of its Jacobian. In the example of the heat emulator, we obtain $K_{\max} = 10$, which reduces the number of physics solver iterations for training by 60%. We denote this second type of reductions as *IC savings*.

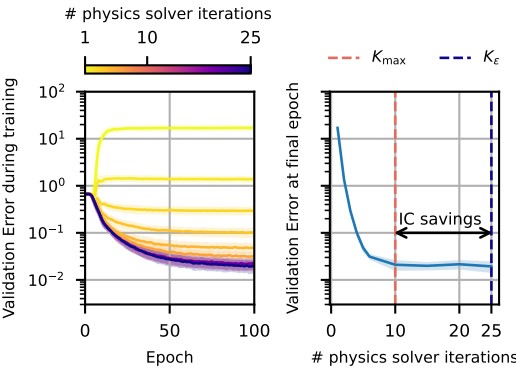

Figure 4: Network accuracy does not improve beyond a refinement level of differentiable physics ($K_{\max}$) significantly lower than full convergence ($K_\epsilon$) constituting *incomplete convergence* (IC) savings.

While we could not isolate a single cause for these IC savings, we hypothesize that it is likely due to a combination of factors: (1) the more critical components of the gradient may converge more rapidly than less important ones, (2) the inherent noisiness of neural network training due to stochastic mini-batching, (3) the usage of momentum-based optimizers, and (4) the inherently approximative nature of machine learning models. Taken together, these factors presumably reduce the need for full convergence in the differentiable physics solver, resulting in very substantial cost savings.

## 3 PROGRESSIVELY REFINED DIFFERENTIABLE PHYSICS

The inner iterations saved by progressive refinement (PR) and incomplete convergence (IC) can greatly reduce training time without impeding accuracy. However, a suitable refinement schedule and $K_{\max}$ are unknown a priori. They depend on the PDE, its discretization (given by $\Lambda$ and $\beta$), the iterative linear solver, and the learning dynamics.

To arrive at a practical method, we automate the progressive refinement and the detection of $K_{\max}$ based on the observation that training progress in terms of a validation metric stagnates when the physics inaccuracy is too large. This stagnation corresponding to different physics refinement levels is exemplified in figures 3 and 4. Conversely, we track training progress over time to automatically increase physics refinement by $\Delta K$ linear solver iterations when training plateaus. Typically PRDP starts at $K_0 = 1$.

---

[2]Throughout this work, we use superscripts in square brackets to denote sequence entries, for example for temporal snapshots or iterates of a linear solver.

```python
def update(model, state_in, state_out, inner_iterations):
    # neural network gives first time step, physics gives second time step
    state_1 = model(state_in)
    state_2 = physics(state_1, inner_iterations)
    loss, grad = value_and_grad(loss_fn)(state_2, state_out)
    model = update_model(model, grad)
    return model, loss

inner_iterations = 1
val_loss_history = [evaluate(model, val_dataloader)]
for epoch in range(epochs):
    for state_in, state_out in dataloader:
        model, loss = update(model, state_in, state_out, inner_iterations)
    val_loss_history.append(evaluate(model, val_dataloader))
    if should_refine(val_loss_history):
        inner_iterations += 1
```

Listing 1: A typical mixed-chain learning pipeline (figure 1) as used in our neural emulator learning experiments. Training begins with coarse physics, using, e.g., 1 solver iteration. The `should_refine` function applies PRDP, determining when to refine and progressively increasing the inner solver iterations during training.

**Controlling Refinement** Given an exponentially smoothed history of validation metrics $\{\tilde{L}_{\mathrm{val}}^{[e]}\}_e$ after a representative training interval, for which we can either use epochs or a fixed number of update steps, we distinguish three trends:

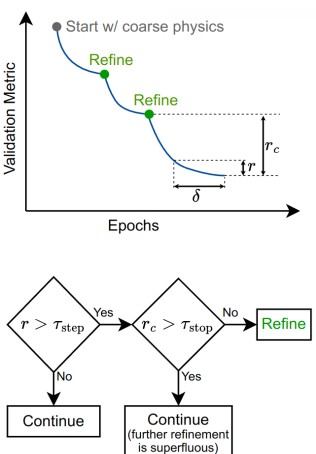

1. **Validation metric plateaus**: If the ratio of the latest validation error and the value of several grace intervals $\delta$ earlier, $r = \tilde{L}_{\mathrm{val}}^{[-1]}/\tilde{L}_{\mathrm{val}}^{[-\delta]}$, is above a threshold $r > \tau_{\mathrm{step}}$, it indicates that the network has achieved the highest possible accuracy at the current refinement level. Then, subject to the following checks, the refinement is increased by $\Delta K$. We record the validation metric before refinement as a checkpoint $c \leftarrow \tilde{L}_{\mathrm{val}}^{[-1]}$. Identifying this trend leads to full utitilization of each refinement level and contributes the PR savings.

2. **Adequate refinement is achieved**: We compute the improvement of the current plateaued validation metric against the checkpoint, $r_c = \tilde{L}_{\mathrm{val}}^{[-1]}/c$. If the stagnation is greater than its threshold, i.e., $r_c > \tau_{\mathrm{stop}}$, the refinement level is retained. This identifies the adequate level of refinement necessary, $K_{\mathrm{max}}$, enabling the IC savings.

3. **Initially divergent regime**: For some scenarios, there is a minimum number of iterations $K_{\mathrm{min}}$ which if $K < K_{\mathrm{min}}$ lead to the convergence of the network to a worse value than its initialization, e.g., for $K = 1$ and $K = 2$ in figure 4. To overcome this, we also refine if $r_c > 1.0$.

Figure 5: Top: the typical training progress of a neural network supported by PRDP, showing the ratios $r$ and $r_c$. Bottom: a simplified flowchart representation of the PRDP control algorithm.

Figure 5 illustrates the basic concept of the PRDP control algorithm. The exact algorithm is detailed in pseudocode in Algorithm 4. Its implementation in a training pipeline that uses differentiable physics is represented by the `should_refine` function in listing 1.

**Applying Refinement** Intrinsically, PRDP pertains to refinement of the primal physics. As we showed in section 2.1, the adjoint refinement is controlled based on the kind of differentiation used. In *implicit differentiation,* an additional linear system solve is performed via a custom differentiation rule. This approach inherently decouples the convergence of the primal and its Jacobian. We maintain their coupling by using the same solver and number of inner iterations for the primal solve and the VJP solve. This setup ensures that adjustments to the number of inner iterations consistently affect the primal and the VJP. When using *unrolled differentiation*, progressive refinement of the primal trivially extends to progressive refinement of the Jacobian. This is because the number of iterations unrolled by the automatic differentiation engine is the same as that of the primal solve.

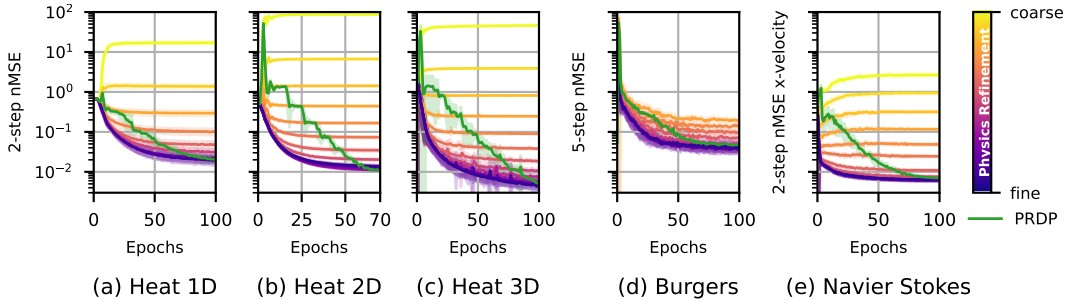

(a) Heat 1D     (b) Heat 2D     (c) Heat 3D     (d) Burgers     (e) Navier Stokes

Figure 6: PRDP achieves the same network accuracy as fully converged physics (baseline in dark purple). It does so by adjusting physics accuracy adaptively over the training if validation performance plateaus, which is particularly noticeable in Heat 2D.

## 4 EXPERIMENTS

To validate our approach across problems of varying complexities, a combination of different learning and physics scenarios are chosen. This includes the convex optimization of an exactly solvable inverse problem with the Poisson equation as described in section 2.2, the learning of autoregressive neural emulators in the spirit of Bar-Sinai et al. (2019) and Brandstetter et al. (2022) for the diffusion and Burgers equation but with a loss setup involving differentiable physics, and third, the learning of a neural-hybrid emulator in a setting similar to Um et al. (2020). When training neural networks, our results show the aggregation over ten different initialization seeds. The validation metric used is the normalized mean squared error (nMSE) over the validation dataset. Further specifics are provided in appendix F.

### 4.1 LINEAR INVERSE PROBLEM

In section 2.2, we demonstrated the potential progressive refinement savings on a doubly-convex inverse Poisson problem when employing a *pre-defined* linear scheduling of inner iterations across outer iterations. This schedule was hand-tuned based on exhaustive runs. PRDP is designed to be problem-independent and adjust the inner iterations adaptively. To confirm this, we apply PRDP in the same setting and achieve a saving of 33% due to progressive refinement. This is slightly higher than the manually achieved 28%.

Moreover, we extended this inverse problem to a three-dimensional parameter space, with each entry scaling the first three eigenmodes, and covered a combination of different setups, including the steepest descent solver (SD), and implicit differentiation (Impl) next to unrolled differentiation (Unr). The results presented in the table to the right show that PRDP universally applies to each combination and is, hence, agnostic to the linear solver

| P | Type | Diff | Cost | Red. | PR. Sav. |
|---|------|------|------|------|----------|
| 1 | Jac | Unr | 75K | 24.87K | 33% |
| 1 | Jac | Impl | 75K | 24.87K | 33% |
| 3 | Jac | Unr | 337.5K | 163K | 48% |
| 3 | Jac | Impl | 337.5K | 163K | 48% |
| 3 | SD | Unr | 192K | 82.7K | 43% |
| 3 | SD | Impl | 192K | 82.7K | 43% |

and the differentiation method. The qualitative behavior of all combinations is displayed in appendix figure 15. For brevity, we only list the PR savings achieved using PRDP by displaying the cost of optimization with $K_\epsilon$ and the corresponding reduction (Red.). Importantly, under the larger parameter space within each of the four setups, PRDP works equally well, achieving PR savings of 48% and 43% for the Jacobi method and the steepest descent solver, respectively.

### 4.2 LINEAR NEURAL EMULATOR LEARNING

We train a multilayer perceptron (outer problem) to function as an autoregressive emulator, i.e., a replacement for the numerical time stepper of a heat diffusion equation. Following the *differential physics* paradigm, the numerical heat equation solver (inner problem) is included in the gradient loop during training. This training pipeline is depicted in figure 1.

In section 2.3, using a 1D case of the same setup, we demonstrated that savings due to *incomplete convergence* are actually achievable, and that they are likely caused by the nature of the training in deep learning. We identified $K_{\max}$ based on an exhaustive search of different $K$ values. This is infea-



Figure 7: Total Inner iters in Heat 1D, 2D & 3D.

sible in practice. Hence, PRDP is designed to automatically find the upper refinement threshold and prevent further superfluous refinement. In figure 6(a), (b) & (c), this can be seen as its convergence follows certain refinement levels. We can confirm that the same $K_{\max} = 10$ is found by PRDP constituting the aforementioned 60% IC savings. Moreover, as figure 7(a) reveals, the total PRDP savings are 79% against the fully converged run with $K_\epsilon = 25$ because PRDP additionally contributes 19% savings due to *progressive refinement*. We repeated a similar experiment in two and three dimensional settings. In the 2D case, the difference between $K_\epsilon = 43$ and $K_{\max} = 12$ constitutes 72% IC savings. Together with 14% PR savings, this totals 86% savings due to applying PRDP as visualized in figure 7. Similarly training a ResNet (He et al., 2016) in the 3D case, the total savings due to PRDP were 81%. This corresponded to a reduction in total training time by 24% and 78% in the 2D and 3D cases, respectively, which underscores PRDP's potential for large computational savings in more difficult, higher-dimensional settings.

## 4.3 NONLINEAR NEURAL EMULATOR LEARNING

The preceding tests constituted inner problems where only the linear system's right-hand side was dependent on the trainable parameters. We expect our PRDP algorithm to work equally well when training through inner problems with a non-constant matrix assembly function $\Lambda$. To illustrate this, we train a ResNet to function as an autoregressive emulator for the one-dimensional Burgers equation on a periodic unit interval with a time-implicit upwinding discretization. Additionally, this yields a non-symmetric matrix requiring the more sophisticated *GMRES* solver (Saad & Schultz, 1986). We use a linearization around the previous point in time, resulting in an Oseen problem (Turek, 1999). When training a ResNet under a similar mixed scenario as in section 4.2, we can again confirm the working of PRDP in this case. It amounts to 30% savings by incomplete convergence reducing $K_\epsilon = 23$ to $K_{\max} = 16$. An additional 29% PR savings

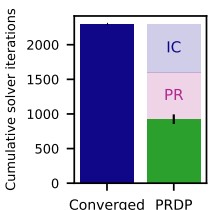

Figure 8: Total inner iters in Burgers.

yield a total of 59% PRDP savings. The Burgers emulation problem was particularly challenging, requiring us to set $K_0 = 4$ to overcome a strongly divergent regime if the physics was too coarse at the beginning of training. This is noticeable in the initial epochs in figure 6(c). This figure also shows that since the fanning between the different (convergent) refinement levels is not as strong as before, the IC savings are lower. However, we still see significant savings due to progressive refinement.

## 4.4 NEURAL-HYBRID EMULATOR FOR THE NAVIER-STOKES EQUATION

Tightly combining classical numerical solvers and neural networks is a promising research direction (Kochkov et al., 2021; Um et al., 2020). It differs from our previous experiments in that the solver is not just part of the training compute graph but will also be executed during the inference of the model.

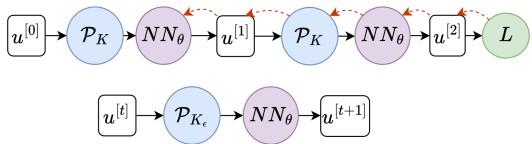

Figure 9: The training (top) and evaluation (bottom) pipelines for neural-hybrid emulator.

We mimic the setup of Um et al. (2020) to have a neural component correct the *discretization error* of an incompressible Navier-Stokes solver by training it against a reference pro-

duced at a higher resolution of discretization. The training and evaluation setups are shown in figure 9. The outer and inner problems correspond to the network training and the Navier-Stokes solver in the training loop, respectively. Similarly to the Burgers example, we choose an upwind-based discretization with the linearization around the previous step in time. The additional incompressibility constraint leads to a saddle point structure (Turek, 1999). We choose to solve it in a coupled form with the GMRES algorithm (Saad & Schultz, 1986).

In this complex scenario, PRDP likewise gives substantial benefits: It is able to save $65\%$ inner iterations due to incomplete convergence and $16\%$ iterations in progressive refinement. Ultimately, reducing the number of iterations by a total of $81\%$ results in a reduction in wall clock training time by $62\%$, as shown in figure 2(c). Since neural-hybrid approaches execute the fully converged physics solver during inference and consequently during validation, PRDP's reliance on a validation metric necessitates this additional cost. In our experiments, we found that compared to training with fully converged physics without computing the validation loss, PRDP was still faster by 57%.

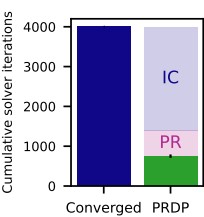

Figure 10: Total inner iters in Navier Stokes.

## 5 RELATED WORK

**Differentiation through implicit relations**  Fischer (1991) established an implementation framework for *unrolled differentiation* and investigated the convergence of the derivative, focusing on linear solvers. Charles (1992) extended this work to a broader class of iterative processes. As an alternative to unrolled differentiation, Christianson (1998) derived the *implicit differentiation* rules over linear solves. Beyond linear systems, implicit differentiation has since gained prominence in the machine learning community, particularly for hyperparameter optimization (Bengio, 2000) and other *bilevel optimization* (BLO) applications that require differentiating through inner optimizations, such as deep equilibrium models (Bai et al., 2019) and meta-learning (Andrychowicz et al., 2016). We view differentiable physics (Thuerey et al., 2021) as another type of BLO with the specialty of sparse linear systems. The practicality of implicit differentiation is largely due to its reduced memory footprint and lower reverse-mode computational cost, especially as modern automatic differentiation (AD) tools can automatically handle the necessary propagation rules (Blondel et al., 2022). Additionally, implicit differentiation allows for the black-box use of solver implementations, enabling the integration of third-party components into differentiable computational graphs (Giles, 2008). However, unrolled differentiation remains an active research area, with work focusing on non-asymptotic analyses (Scieur et al., 2022) and other aspects (Maclaurin et al., 2015; Franceschi et al., 2017; Grazzi et al., 2020; Ji et al., 2021).

**Analysis and cost-reduction of bilevel optimization**  To address the computational cost of bilevel optimization, Fung et al. (2021) presented *Jacobian-free backpropagation*, where the Jacobian of the implicit solver is approximated as the identity, eliminating the need for adjoint linear solves. Geng et al. (2021) introduced *phantom gradients*, where the matrix inversion is replaced with an approximate inverse, and provided theoretical guarantees on the convergence of stochastic gradient descent as the outer problem. Lorraine et al. (2020) approximates the implicit linear solves with a reduced number of conjugate gradient steps. Moreover, Shaban et al. (2019) and recently Bolte et al. (2023) discuss unrolled differentiation through only a reduced number of iterations. All the aforementioned works consider (what we call) *incomplete convergence* (IC) savings, albeit in the setting of deep equilibrium models (Bai et al., 2019) or hyperparameter optimization (Feurer & Hutter, 2019). On the other hand, Pedregosa (2016) studied approximate hypergradients by adjusting the tolerances of inner primal and implicit linear solves following a pre-defined schedule. This work proves (what we call) *progressive refinement* (PR) savings. Prior to this, Domke (2012) investigated outer training with unrolled AD through incompletely converged iterations, specifically for gradient descent, heavy ball, and L-BFGS methods as inner optimizers. They note the advantage of implementing incomplete convergence using the number of inner iterations rather than inner tolerance. Our approach uniquely combines PR and IC savings – through both unrolled and implicit differentiation – targeting iterative solvers for sparse linear systems embedded within neural network training. To our knowledge, no prior work has applied these techniques in this context.

## 6    LIMITATIONS AND OUTLOOK

**Limitations**    While PRDP is designed to work across a range of differentiable physics settings, there are, of course, way more potential linear systems that can arise, all with their specific characteristics. Albeit we believe that the approach using scheduling over iterations rather than scheduling via tolerances might be more generally applicable, it remains to be tested how PRDP applies to unstructured discretizations in higher dimensions potentially also involving multiple physics. PRDP is limited to settings that involve iterative linear solvers. As such, it can not be used for purely explicit numerical solvers (e.g., found in strongly hyperbolic problems) or when the linear systems are solved spectrally (Kochkov et al., 2024). However, most other simulations in science and engineering entail linear solvers either due to more efficient implicit time stepping or when solving steady-state problems. Moreover, whenever dynamics are constrained, e.g., the incompressible Navier-Stokes equations, even if purely explicit time stepping is used, iterative processes are required to resolve the constraints.

**Outlook and Impact**    Progressively Refined Differentiable Physics (PRDP) provides a means to exploit both savings due to *progressive refinement* and *incomplete convergence*, thereby greatly reducing the cost of neural network training with differentiable physics. This could enable settings that so far have been infeasible due to prohibitive expenses like long temporal unrollings or differentiable physics on high resolutions or in three dimensions.

Our work opens up many interesting directions for future investigations, such as smoother relations between the achieved plateau ratio $r$ and the conducted iterations/prescribed tolerance in the physics solver. Potentially, those could be faster than the linear increments we used in this work. For neural-hybrid emulators, IC savings via $K_{\text{max}}$ could also extend to the inference stage. So far, PRDP couples primal and adjoint (in-)accuracy. One can use the unique properties of either unrolled or implicit differentiation for more sophisticated approaches. This can include an imbalanced number of iterations in the primal solve and the implicit linear solve. For unrolled differentiation, one can unroll a different number of iterations either reversely following Bolte et al. (2023) or Shaban et al. (2019) or from the beginning. Other levels of refinement than by the number of iterations of a linear solver are likewise interesting directions for future work, e.g., using differences in spatial or temporal resolution together with resolution-agnostic neural emulators.

## 7    CONCLUSION

This work investigated neural network training through incompletely converged differentiable physics. Our objective was to reduce the cost of gradient computation without sacrificing training accuracy. Prevalent research on training with incompletely converged gradients focuses on bilevel optimization problems, primarily in meta-learning or hyperparameter optimization contexts. This work extends the research in the differentiable physics space, focusing on iterative linear system solves associated with discretized differential operators.

We have demonstrated that neural networks can be trained through differentiable physics solvers significantly coarser than full convergence. Our approach of Progressively Refined Differentiable Physics combines compute savings from both progressive refinement and incomplete convergence. This yielded favorable outcomes across all our test scenarios. It makes initial training progress cheap using coarse physics and carefully improves training accuracy using adaptive physics refinement over time, ending the training at a refinement significantly below primal convergence. In total, we achieve up to $86\%$ fewer cumulative number of physics solver iterations than training with fully converged physics, and $78\%$ reduction in training time. Our approach has the potential for numerous practical improvements in learning pipelines that involve differentiable numerical solvers and could facilitate integrating simulators into training that were previously considered computationally infeasible.

## 8    REPRODUCIBILITY STATEMENT

To ensure reproducibility, we detail all physics parameters, discretization schemes, boundary conditions, iterative solvers, network architectures, optimizers, learning rate schedules, data generation

methods, train-test splits, and batch sizes in the appendix. Additionally, the full source code for our experiments is available at `https://github.com/tum-pbs/PRDP`.

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

# A NOTATION

This section lists the primary notations used throughout the paper for clarity and ease of reference.

- $\Omega \subset \mathbb{R}^D$: Spatial domain of the physics problem.
- $h$: Spatial discretization width.
- $t$: Time step index in time-stepping problems.

VARIABLES AND OPERATORS

- $\boldsymbol{u}_h \in \mathbb{R}^N$: The discretized solution vector.
- $\boldsymbol{u}_h^*$: The direct solution to the linear system $\boldsymbol{A}\boldsymbol{u}_h = \boldsymbol{b}_h$.
- $\boldsymbol{A} \in \mathbb{R}^{N \times N}$: The system matrix of the discretized PDE.
- $\boldsymbol{b}_h \in \mathbb{R}^N$: The right-hand side vector.
- $\Delta$: The Laplacian operator, e.g., $\Delta u(x)$.
- $\nabla$: The gradient operator.
- $\lambda$: Auxiliary variable used in implicit differentiation.
- $\beta(\theta)$: Function mapping parameters $\theta$ to $\boldsymbol{b}_h$.
- $\Lambda(\theta)$: Funcion mapping parameters $\theta$ to $\boldsymbol{A}$.
- $\Phi(.)$: Function representing each iteration of an iterative solver.
- $\| \cdot \|$: Matrix or vector norm.
- $f(.; \theta)$: Neural network parameterized by $\theta$.
- $.^T$: Transpose of a matrix or vector.
- $\vec{1}$: Identity matrix.
- $\theta$: Trainable parameters of the neural network or physical model.
- $K$: Number of iterations performed by the iterative solver.
- $K_\epsilon$: Number of solver iterations required to achieve tolerance $\epsilon$.
- $\mathcal{P}_K$: Differentiable physics operator, approximating the solution of a linear system after $K$ iterations.
- $\boldsymbol{u}_h^{[K]}$: Approximate solution of the linear system after $K$ iterations of the solver.
- $\boldsymbol{u}_h^r$: Reference solution to the linear system, computed using either a direct solver or a fully converged iterative solver.
- $L(\theta)$: Outer loss function, which measures the discrepancy between the predicted solution and a reference solution.
- $\boldsymbol{J}_l, \boldsymbol{J}_{\mathcal{P}_K}$: Jacobians of the loss function and the physics operator, respectively.

PARAMETERS AND TOLERANCES

- $\epsilon$: Convergence tolerance for iterative solvers.
- $\tau_{\text{step}}$: Threshold for the stepping criterion in PRDP.
- $\tau_{\text{stop}}$: Threshold for the stopping criterion in PRDP.
- $\delta$: Grace window for epoch intervals.
- $e$: Exponential averaging window
- $\Delta K$: Increment in solver iterations used in progressive refinement.

MISCELLANEOUS

- $K_\epsilon$: Number of iterations required to reach a specified tolerance $\epsilon$ in the iterative solver.

- $K_{\max}$: Maximum number of inner iterations sufficient for training accuracy.

- $K_{\min}$: Minimum number of inner iterations required to prevent divergence during training.

- $r, r_c$: Ratios used to evaluate the stepping and stopping criteria for progressive refinement, based on the validation metric's behavior.

## B  ITERATIVE LINEAR SOLVERS

The solution to linear systems of equations is fundamental to scientific computing. Especially for partial differential equations discretized using fine resolutions or in higher dimensions, the discrete linear systems become large and sparse. Oftentimes, iterative solvers are the only practical way of solving them (Saad, 2003).

For our efforts to reduce the cost of differentiable physics as part of neural network training, we consider three different iterative solvers. The *Jacobi method* belongs to the class of smoothing/relaxation methods. When reformulating a linear system solve as a convex quadratic optimization problem, the algorithm of *steepest descent* naturally arises. To solve asymmetric and complicated systems, we also use the more sophisticated *GMRES* method. We implemented the Jacobi method and steepest ourselves. For the GMRES, we used the version of JAX[1].

In all tests, the linear solvers were zero-initialized. Convergence is achieved if the relative residuum error using the 2-norm

$$\xi^{[k]} = \frac{\|\boldsymbol{A}\boldsymbol{u}_h^{[k]} - \boldsymbol{b}_h\|_2}{\|\boldsymbol{b}_h\|_2} \tag{6}$$

is below the threshold $\epsilon$, which we set to $\epsilon = 10^{-5}$ due to single precision. We use the maximum number of iterations $K$ as a way to control the refinement of the physics simulation. The iterative solvers return if either the maximum number of iterations $K$ is reached or the tolerance threshold $\xi^{[k]} < \epsilon$ is met.

### B.1  JACOBI RELAXATION

The Jacobi method is a relaxation-type method based on the decomposition of the system matrix $\boldsymbol{A}$ into a strictly lower diagonal $\boldsymbol{L}$, a diagonal $\boldsymbol{D}$, and a strictly upper diagonal $\boldsymbol{U}$ part such that $\boldsymbol{A} = \boldsymbol{L} + \boldsymbol{D} + \boldsymbol{U}$. We present its algorithm in 1.

---

**Algorithm 1:** Jacobi Method for Solving $\boldsymbol{A}\boldsymbol{u}_h = \boldsymbol{b}_h$

---

**Input:** Matrix $\boldsymbol{A} \in \mathbb{R}^{N \times N}$, vector $\boldsymbol{b}_h \in \mathbb{R}^n$, tolerance $\epsilon$, maximum iterations $K$
**Output:** Approximation of the solution $\boldsymbol{u}_h \in \mathbb{R}^N$
Decompose $\boldsymbol{L} + \boldsymbol{D} + \boldsymbol{U} = \boldsymbol{A}$ Initialize $\boldsymbol{u}_h^{[0]} \in \mathbb{R}^N$ to zeros $\boldsymbol{u}_h^{[0]} = \boldsymbol{0}$;
**for** $k = 0, 1, \ldots, K-1$ **do**
  $\boldsymbol{u}_h^{[k+1]} = \boldsymbol{D}^{-1}\left(\boldsymbol{b}_h - (\boldsymbol{L} + \boldsymbol{D})\boldsymbol{u}_h^{[k]}\right)$
  Compute relative residuum norm: $\xi^{[k+1]} = \|\boldsymbol{A}\boldsymbol{u}_h^{[k+1]} - \boldsymbol{b}_h\|_2/\|\boldsymbol{b}_h\|_2$;
  **if** $\xi^{[k+1]} < \epsilon$ **then**
    **break**;
  **end**
**end**
**return** $\boldsymbol{u}_h^{[k+1]}$

---

[1] https://jax.readthedocs.io/en/latest/_autosummary/jax.scipy.sparse.linalg.gmres.html

As a smoothing method, the convergence of the Jacobi method depends on the spectral radius of the iterator matrix (Saad, 2003) via

$$\|\boldsymbol{r}_h^{[k+1]}\|_2 \leq \rho(\boldsymbol{D}^{-1}(\boldsymbol{L} + \boldsymbol{R}))\|\boldsymbol{r}_h^{[k]}\|_2. \tag{7}$$

Loosely speaking, the condition number of the system matrix $\boldsymbol{A}$ affects the spectrum of the Jacobi iterator matrix in that high condition numbers lead to spectral radii close to 1.0, causing slow convergence.

## B.2 STEEPEST DESCENT

The steepest descent method follows the gradient of the convex quadratic optimization problem associated with solving a linear system of equations, which is also given by the negative residuum. However, in contrast to the gradient descent methods typically found in more general optimization problems (like for training neural networks), the optimal step size for maximum decrease $\alpha^{[k]}$ can be determined in each iteration (Saad, 2003). We present the steepest descent in algorithm 2.

---

**Algorithm 2:** Steepest Descent Method for Solving $\boldsymbol{A}\boldsymbol{u}_h = \boldsymbol{b}_h$

---

**Input:** Matrix $\boldsymbol{A} \in \mathbb{R}^{N \times N}$, vector $\boldsymbol{b}_h \in \mathbb{R}^N$, tolerance $\epsilon$, maximum iterations $K$
**Output:** Approximation of the solution $\boldsymbol{u}_h \in \mathbb{R}^N$
Initialize $\boldsymbol{u}_h^{[0]} \in \mathbb{R}^N$ to zeros $\boldsymbol{u}_h^{[0]} = \boldsymbol{0}$;
Set initial residual $\boldsymbol{r}_h^{[0]} = \boldsymbol{b}_h - \boldsymbol{A}\boldsymbol{u}_h^{[0]}$;
**for** $k = 0, 1, \ldots, K - 1$ **do**
  Compute step size $\alpha^{[k]} = \frac{\boldsymbol{r}_h^{[k]} \cdot \boldsymbol{r}_h^{[k]}}{\boldsymbol{r}_h^{[k]} \cdot \boldsymbol{A}\boldsymbol{r}_h^{[k]}}$;
  Update solution $\boldsymbol{u}_h^{[k+1]} = \boldsymbol{u}_h^{[k]} + \alpha^{[k]}\boldsymbol{r}_h^{[k]}$;
  Update residual $\boldsymbol{r}_h^{[k+1]} = \boldsymbol{r}_h^{[k]} - \alpha^{[k]}\boldsymbol{A}\boldsymbol{r}_h^{[k]}$;
  Compute relative residuum norm: $\xi^{[k+1]} = \|\boldsymbol{r}_h^{[k+1]}\|_2 / \|\boldsymbol{b}_h\|_2$;
  **if** $\xi^{[k+1]} < \epsilon$ **then**
    **break**;
  **end**
**end**
**return** $\boldsymbol{u}_h^{[k+1]}$

---

It can be shown that the residuum norm converges exponentially linear in the asymptotic regime based on the condition number of the system matrix $\kappa(\boldsymbol{A})$ via (Saad, 2003)

$$\|\boldsymbol{r}_h^{[k+1]}\|_2 \leq \left(\frac{\kappa(\boldsymbol{A}) - 1}{\kappa(\boldsymbol{A}) + 1}\right)\|\boldsymbol{r}_h^{[k]}\|_2. \tag{8}$$

## B.3 GMRES

The General Method of RESiduals (GMRES) builds a subset of the Krylov basis associated with system matrix $\boldsymbol{A}$ and finds an approximation to the solution of the linear system via least-squares. Typically, it is restarted to rebuild a new Krylov basis each $m$ iteration. We call an iteration $k$ the construction of the entire $m$-dimensional Krylov subspace and subsequent least-squares solve. We use the `"batched"` mode of the JAX implementation, which can only terminate after a restart but not within a restart.

## C MORE DETAILS ON DIFFERENTIATING OVER ITERATIVE LINEAR SOLVERS

For convenience, we restate the simple computational graph involving a linear solver with $K$ steps subject to reverse-mode automatic differentiation (AD) as

$$(\nabla_\theta L)^T = \bar{\theta}^T = \left(\left(\vec{1}^T \boldsymbol{J}_l|_{\mathcal{P}_K(\boldsymbol{f}(\cdot\,;\theta))}\right) \boldsymbol{J}_{\mathcal{P}_K}|_{\boldsymbol{f}(\cdot\,;\theta)}\right) \boldsymbol{J}_{\boldsymbol{f}}|_\theta. \tag{9}$$

---

**Algorithm 3:** GMRES Method for Solving $\boldsymbol{A}\boldsymbol{u}_h = \boldsymbol{b}_h$

---

**Input:** Matrix $\boldsymbol{A} \in \mathbb{R}^{N \times N}$, vector $\boldsymbol{b}_h \in \mathbb{R}^N$, tolerance $\epsilon$, maximum iterations $K$, restart
      parameter $m$

**Output:** Approximation of the solution $\boldsymbol{u}_h \in \mathbb{R}^N$

Initialize $\boldsymbol{u}_h^{[0]} \in \mathbb{R}^N$ to zeros $\boldsymbol{u}_h^{[0]} = \boldsymbol{0}$;

Set initial residual $\boldsymbol{r}_h^{[0]} = \boldsymbol{b}_h - \boldsymbol{A}\boldsymbol{u}_h^{[0]}$;

Set $\beta = \|\boldsymbol{r}_h^{[0]}\|_2$;

Set $\boldsymbol{v}_1 = \boldsymbol{r}_h^{[0]}/\beta$ (first Krylov basis vector);

**for** $k = 0, 1, \ldots, K-1$ **do**

    **for** $j = 1, 2, \ldots, m$ **do**

        Compute $\boldsymbol{w}_j = \boldsymbol{A}\boldsymbol{v}_j$;

        **for** $i = 1, \ldots, j$ **do**

            Compute $h_{ij} = \boldsymbol{v}_i \cdot \boldsymbol{w}_j$;

            Update $\boldsymbol{w}_j = \boldsymbol{w}_j - h_{ij}\boldsymbol{v}_i$;

        **end**

        Set $h_{j+1,j} = \|\boldsymbol{w}_j\|_2$;

        Normalize $\boldsymbol{v}_{j+1} = \boldsymbol{w}_j/h_{j+1,j}$;

    **end**

    Solve the least-squares problem: minimize $\|\beta\boldsymbol{e}_1 - \boldsymbol{H}^{[k]}\boldsymbol{y}^{[k]}\|_2$;

    Update solution: $\boldsymbol{u}_h^{[k+1]} = \boldsymbol{u}_h^{[k]} + \boldsymbol{v}^{[k]}\boldsymbol{y}^{[k]}$;

    Compute relative residuum norm: $\xi^{[k+1]} = \|\boldsymbol{r}_h^{[k+1]}\|_2/\|\boldsymbol{b}_h\|_2$;

    **if** $\xi^{[k+1]} < \epsilon$ **then**

        **break**;

    **end**

    Restart GMRES after every $m$ iterations if convergence not met;

**end**

**return** $\boldsymbol{u}_h^{[k+1]}$

---

In this setting, one has to compute three vector-Jacobian products (VJPs). The VJP over the loss function $\bar{\boldsymbol{u}}_h^T = \bar{\mathbf{1}}^T \, \boldsymbol{J}_l|_{\mathcal{P}_K(\boldsymbol{f}(\cdot\,;\theta))}$ and into the network's parameter space $\bar{\theta}^T = \bar{\boldsymbol{g}}_h^T \boldsymbol{J}_f|_\theta$ can be straightforwardly evaluated by the AD engine as they consist of explicit operations.

On the other hand, the VJP over the approximative solver $\bar{\boldsymbol{g}}_h^T = \bar{\boldsymbol{u}}_h^T \boldsymbol{J}_{\mathcal{P}_K}|_{\boldsymbol{f}(\cdot\,;\theta)}$ is nontrivial as it requires handling the iterative solver and the assembly routines. We detail the two prominent approaches below. They are visually depicted in figure 11.

### C.1 IMPLICIT DIFFERENTIATION

We first solve an auxiliary linear system for an adjoint variable $\lambda_h$ (Christianson, 1998)

$$\boldsymbol{A}^T \lambda_h = \bar{\boldsymbol{u}}_h. \tag{10}$$

Since the convergence behavior of most iterative linear solvers is dependent on the spectrum of the system matrix and since transposition does not effectively change this, it is reasonable to employ the same iterator as in the primal solution but with a transposed system matrix and different right-hand side $\Phi(\cdot\,;\boldsymbol{A}^T, \bar{\boldsymbol{u}}_h)$. The convergence of the iterates $\{\lambda_h^{[0]}, \lambda_h^{[1]}, \lambda_h^{[2]}, \ldots, \lambda_h^{[\tilde{K}]}\} = \{\Phi^k(\lambda_h^{[0]}; \boldsymbol{A}^T, \bar{\boldsymbol{u}}_h)\}_{k=0}^{\tilde{K}}$ can potentially be different than in the primal solution. In other words, $\tilde{K}_\epsilon$ for tolerance $\epsilon$ can be different from primal the $K_\epsilon$.

Once $\lambda_h$ is determined, it directly equals the intermediate gradient on the right-hand side $\bar{\boldsymbol{b}}_h = \lambda_h$. The intermediate gradient on the system matrix arises as the negative outer product with the *primal* solution $\bar{\boldsymbol{A}} = -\lambda_h \boldsymbol{u}_h^T$. Hence, it is sufficient to only save the solution $\boldsymbol{u}_h^{[K]}$ from the primal pass; no further iterates are required.

Then, the reversely propagated intermediate gradient on the input of the physics operator is given by

$$\bar{u}_h^T \boldsymbol{J}_{\mathcal{P}_K}|_{\boldsymbol{g}_h} = \bar{\boldsymbol{g}}_h^T = \bar{\boldsymbol{b}}_h^T \boldsymbol{J}_\beta|_{\boldsymbol{g}_h} + \underbrace{\bar{\boldsymbol{A}}^T \boldsymbol{J}_\Lambda|_{\boldsymbol{g}_h}}_{=\bar{\boldsymbol{A}}:\boldsymbol{J}_\Lambda|_{\boldsymbol{g}_h}}, \tag{11}$$

under the abuse of notation to consider the gradient matrix $\bar{\boldsymbol{A}}$ as a vector. Alternatively, this can also be expressed as the left double contraction. It must be noted that typically $\bar{\boldsymbol{A}}$ is only evaluated on the sparsity pattern of $\boldsymbol{A}$, if it is materialized at all.

In case only the right-hand side $\boldsymbol{b}_h$ is parameterized, there will be no reverse propagation through the matrix assembly $\Lambda$. For example, this was the case for the Poisson and heat emulator examples in section 2.2 and section 2.3, respectively.

Since we use matrix-free implementations for most linear solvers, the primal system is not solved with a materialized matrix $\boldsymbol{A}$ but a function that is linear in its first argument

$$\Lambda(\boldsymbol{g}_h)\boldsymbol{u}_h \overset{\triangle}{=} \psi(\boldsymbol{u}_h; \boldsymbol{g}_h). \tag{12}$$

In this case, we compute the system matrix' contribution to the previous intermediate gradient via the *negative* VJP over its second argument

$$\bar{u}_h^T \boldsymbol{J}_{\mathcal{P}_K}|_{\boldsymbol{g}_h} = \bar{\boldsymbol{g}}_h^T = \bar{\boldsymbol{b}}_h^T \boldsymbol{J}_\beta|_{\boldsymbol{g}_h} - \lambda_h^T \boldsymbol{J}_{\psi,2}|_{\boldsymbol{g}_h}. \tag{13}$$

To employ the iterator $\Phi$ on a matrix-free version of the transposed matrix, we need to programmatically transpose the function $\psi$, which can be done with JAX (Bradbury et al., 2018). Moreover, there is a function in JAX to automatically register linear solvers with correct propagation rules[2].

More generally speaking, implicit differentiation is powerful because it allows differentiation over various implicit relations simply by solving a linear system of equations. Modern AD engines allow the effortless linearization of optimality conditions. This allows for easily registering custom propagation rules by employing matrix-free Krylov solvers (Blondel et al., 2022). Note, however, that the focus of this paper is the differentiation over linear system solutions in which the primal operation is the *same* as in the implicit propagation rule.

---

[2] https://jax.readthedocs.io/en/latest/_autosummary/jax.lax.custom_linear_solve.html

### C.2 Unrolled Differentiaion

Unrolled differentiation of an iterative program is the direct application of standard automatic differentiation tools to its algorithmic implementation. AD unrolls the program's iterations and writes them as individual computational steps.

We find the intermediate gradients on both the system matrix $\bar{A}$ and right-hand side $\bar{b}_h$ via first computing the intermediate gradient on all iterates using the VJP of the iterator $\Phi$ with respect to its first argument

$$\left(\bar{u}_h^{[k]}\right)^T = \bar{u}_h^T \prod_{l=K-1}^{k} J_{\Phi,1}\big|_{u_h^{[l]}=\Phi^l(u_h^{[0]};A,b_h)}. \tag{14}$$

Then we can aggregate each contribution using the VJP over the iterator $\Phi$ with respect to its conditioned arguments

$$\bar{A}^T \stackrel{\text{unroll}}{=} \sum_{k=0}^{K-1} \left(\bar{u}_h^{[k]}\right)^T J_{\Phi,2}\big|_{u_h^{[k]}=\Phi^k(u_h^{[0]};A,b_h)}, \tag{15}$$

$$\bar{b}_h^T \stackrel{\text{unroll}}{=} \sum_{k=0}^{K-1} \left(\bar{u}_h^{[k]}\right)^T J_{\Phi,3}\big|_{u_h^{[k]}=\Phi^k(u_h^{[0]};A,b_h)}. \tag{16}$$

Since the VJP of the iterator has to be evaluated at primal inputs, typically, all iterates must be stored on the tape or recomputed. There are approaches that balance compute and memory (Griewank, 1992). However, for simplicity, we only implement unrolled differentiation by retaining the entire sequence of iterates.

After the intermediate gradients have been obtained, the gradients are further backpropagated via equation 11. While the mathematical description of unrolled differentiation is more elaborate than for implicit differentiation, its implementation in AD engines like JAX is easier given the algorithm fully uses differentiable operations.

The non-asymptotic study of Scieur et al. (2022) revealed that the convergence of unrolled differentiation can exhibit a burn-in phenomenon. We also investigated this for our iterative linear solvers but found it to be practically irrelevant. While it theoretically can occur for problems with parameterized system matrices or for Krylov methods in general, we believe that for large enough parameter spaces, the potential burn-in of some gradient components is compensated by the entirety. Moreover, since we applied PRDP using both unrolled and implicit differentiation and observed almost identical savings in both cases, we further conclude that even if there was a burn-in, the PRDP approach would be unaffected by it.

## D Differentiable Solvers

In this section, we describe the discretization choices behind the differentiable solvers. They are based on finite difference approximations on uniform cartesian grids with Dirichlet or periodic boundary conditions. Linear PDEs naturally lead to a linear system of equations for which only the right-hand assembly remains a dependency on the prior compute graph, e.g., is built upon information previous in time. In the case of the nonlinear PDEs, we choose a Picard-based approximation leading to Oseen-like problems (Turek, 1999). Those have both the right-hand side as well as the system matrix depending on the prior parts of the compute graph.

The discretizations are implemented matrix-free in JAX (Bradbury et al., 2018). Matrices are only materialized when needed for smoothing methods or for direct decomposition-based solvers.

Once fully discretized and re-formulated, each problem is cast in the standard form of a linear system of equations

$$Au_h = b_h, \tag{17}$$

using the assembly routines $A = \Lambda(g_h)$ and $b_h = \beta(g_h)$. In the simplest case we have directly $g_h = \theta$ but other setups like information from a previous time step $g_h = u_h^{[t-1]}$ are common. Then, we define the physics operator $\mathcal{P}$ as the function mapping from the prior point in the compute graph

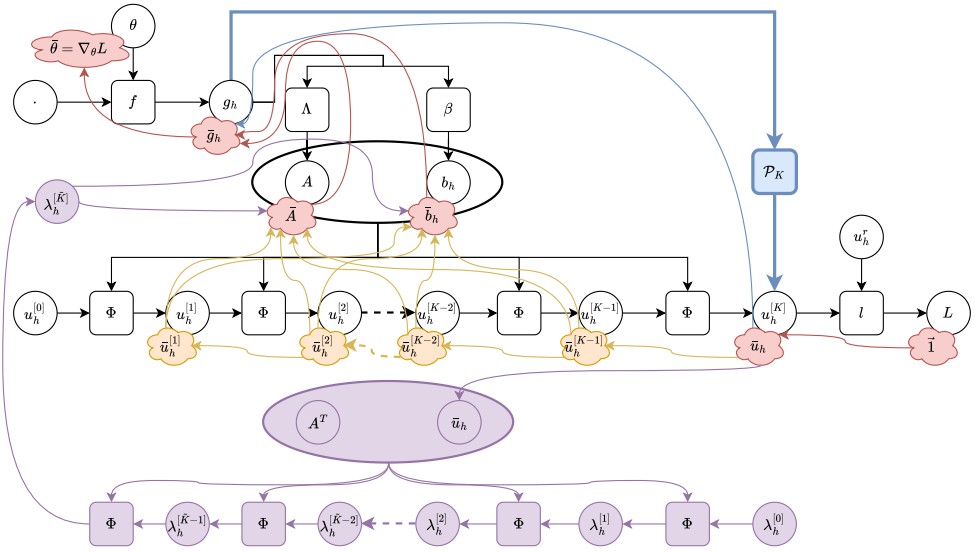

Figure 11: The iterative solution of a linear system of equations spawns a long sequential compute graph that constitutes the primal pass (in black). The step from prior variable $\boldsymbol{g}_h$ to approximate solution $\boldsymbol{u}_h^{[K]}$ can be capsuled as the physics operator $\mathcal{P}_K$ (in blue). Under reverse-mode AD, differentiation over $\mathcal{P}_K$ (blue curvy line) can be done in two ways. *Implicit Differentiation* spawns an auxiliary iterative linear solve using the transposed matrix $\boldsymbol{A}^T$ (in purple). On the other hand, *unrolled differentiation* opens the black box and reversely propagates through each iterator step $\Phi$ (in yellow). Both approaches yield intermediate gradients on system matrix $\bar{\boldsymbol{A}}$ and $\bar{\boldsymbol{b}}_h$, which are subsequently backpropagated into $\bar{\boldsymbol{g}}_h$ via the VJP over their assembly routines.

$\boldsymbol{g}_h$ to the solution of the linear system of equations

$$\mathcal{P}(\boldsymbol{g}_h) := \boldsymbol{u}_h = \boldsymbol{A}^{-1}\boldsymbol{b}_h = (\Lambda(\boldsymbol{g}_h))^{-1}\beta(\boldsymbol{g}_h) \tag{18}$$

If $\mathcal{P}$ is given without subscript this refers to an exact solver of the linear system. This can be a direct method based on matrix decomposition or a fully converged iterative solver. Either way, the residuum norm of this solver's result is guaranteed to be below the relative tolerance threshold $\epsilon$ via

$$\frac{\|\Lambda(\boldsymbol{g}_h)\mathcal{P}(\boldsymbol{g}_h) - \beta(\boldsymbol{g}_h)\|_2}{\|\beta(\boldsymbol{g}_h)\|_2} < \epsilon. \tag{19}$$

If not mentioned otherwise, we set $\epsilon = 10^{-5}$ because we exclusively work in single precision floating format. An approximate solver with $K$ iterations is written $\mathcal{P}_K(\cdot)$. The iterative linear solvers we employed are introduced in section B.

### D.1   POISSON EQUATION IN 1D

We solve the Poisson equation on the unit interval with homogeneous Dirichlet boundary conditions and a parameterized right-hand side

$$\frac{\mathrm{d}^2 u(x)}{\mathrm{d}x^2} = -p(x, \theta) \qquad u(0) = 0 = u(1) \tag{20}$$

For a finite difference discretization, the domain $\Omega = (0, 1)$ is divided equidistantly into $N + 2$ grid points, out of which two are related to the prescribed value on the boundary and can, hence, be ignored. As such, there are $N$ degrees of freedom that make up the discrete solution vector $\boldsymbol{u}_h \in \mathbb{R}^N$. Equally, the right-hand side is discretized at the same points and negated, yielding $\boldsymbol{b}_h$. At index $i$ interior to the domain, the second derivative is approximated with the three-point stencil

$$\frac{1}{(\Delta x)^2}(u_{i-1} - 2u_i + u_{i+1}) = -b_i, \tag{21}$$

with the spacing $\Delta x = \frac{1}{N+1}$. This leads to the system of linear equations

$$
\underbrace{\frac{1}{(\Delta x)^2} \begin{bmatrix} -2 & 1 & 0 & \ldots & 0 & 0 \\ 1 & -2 & 1 & \ldots & 0 & 0 \\ 0 & 1 & -2 & \ldots & 0 & 0 \\ \vdots & & \ddots & & & \vdots \\ 0 & 0 & 0 & \ldots & 1 & -2 \end{bmatrix}}_{=:\tilde{\mathcal{L}}_1 = \boldsymbol{A}} \begin{bmatrix} u_1 \\ u_2 \\ u_3 \\ \vdots \\ u_N \end{bmatrix} = \begin{bmatrix} b_1 \\ b_2 \\ b_3 \\ \vdots \\ b_N \end{bmatrix},
\tag{22}
$$

in which the first matrix is due to the discretized Laplace operator in one dimension. We denote it $\tilde{\mathcal{L}}_1$ (the tilde is to distinguish it from the Laplacian matrices on periodic boundaries of the following sections). The subscript is to indicate the one-dimensional setting. It is tridiagonal and solely defines the system to be solved with $\boldsymbol{A} = \tilde{\mathcal{L}}_1$ as

$$
\boldsymbol{A}\boldsymbol{u}_h = \beta(\boldsymbol{b}_h).
\tag{23}
$$

with the right-hand side assembly function just element-wise negating the input. The system matrix is not parameter-dependent, i.e., $\Lambda(\boldsymbol{g}) = \tilde{\mathcal{L}}_1$.

## D.2 HEAT DIFFUSION IN 1D

We consider the time-dependent diffusion equation in one dimension on the unit interval under homogeneous Dirichlet boundary conditions

$$
\frac{\partial u}{\partial t} = \nu \frac{\partial^2 u}{\partial x^2} \qquad u(t,0) = 0 = u(t,1).
\tag{24}
$$

Similarly to section D.1, we equidistantly divide the domain into $N + 2$ grid points. The matrix associated with the discretized second derivative in one dimension is again

$$
\tilde{\mathcal{L}}_1 := \frac{1}{(\Delta x)^2} \begin{bmatrix} -2 & 1 & 0 & \ldots & 0 & 1 \\ 1 & -2 & 1 & \ldots & 0 & 0 \\ 0 & 1 & -2 & \ldots & 0 & 0 \\ \vdots & & \ddots & & & \vdots \\ 1 & 0 & 0 & \ldots & 1 & -2 \end{bmatrix}.
\tag{25}
$$

With it equation 24 can be discretized in space via the method of lines

$$
\frac{\mathrm{d}\boldsymbol{u}_h}{\mathrm{d}t} = \nu\mathcal{L}_1\boldsymbol{u}_h.
\tag{26}
$$

Applying an implicit Euler discretization to the time derivative yields

$$
\frac{\boldsymbol{u}_h^{[t+1]} - \boldsymbol{u}_h^{[t]}}{\Delta t} = \nu\tilde{\mathcal{L}}_1\boldsymbol{u}_h^{[t+1]},
\tag{27}
$$

with two subsequent time levels on the state vector. This can be rearranged into the standard form of a linear system of equations as

$$
\underbrace{\left(\boldsymbol{I} - \nu\Delta t\tilde{\mathcal{L}}_1\right)}_{:=\boldsymbol{A}} \boldsymbol{u}_h^{[t+1]} = \boldsymbol{u}_h^{[t]}.
\tag{28}
$$

This is also referred to as the backward-in-time central-in-space (BTCS) discretization of the diffusion equation. Advancing the state by one time level requires solving a linear system with the constant system matrix $\boldsymbol{A}$ and with the assembly function $\beta(\cdot)$ simply being the identity. Hence, the physics operator maps to the next state in time with

$$
\boldsymbol{u}_h^{[t+1]} = \mathcal{P}(\boldsymbol{u}_h^{[t]}).
\tag{29}
$$

### D.3 HEAT DIFFUSION IN 2D

The equation for heat diffusion in two dimensions on the unit square with homogeneous Dirichlet boundary conditions reads

$$\frac{\partial u}{\partial t} = \nu \underbrace{\left( \frac{\partial^2 u}{\partial x^2} + \frac{\partial^2 u}{\partial y^2} \right)}_{\Delta u} \qquad u(t,0,y) = 0 = u(t,1,y), u(t,x,0) = 0 = u(t,x,1) \qquad (30)$$

The Laplacian matrix in two dimensions can be written in a block structure using the one-dimensional Laplacian $\tilde{\mathcal{L}}_1$ and appropriately sized identity matrices $\boldsymbol{I}$

$$\tilde{\mathcal{L}}_2 = \frac{1}{(\Delta x)^2} \begin{bmatrix} (\Delta x)^2 \tilde{\mathcal{L}}_1 - 2\boldsymbol{I} & \boldsymbol{I} & \boldsymbol{0} & \boldsymbol{0} & \dots & \boldsymbol{0} & \boldsymbol{0} & \boldsymbol{0} \\ \boldsymbol{I} & (\Delta x)^2 \tilde{\mathcal{L}}_1 - 2\boldsymbol{I} & \boldsymbol{I} & \boldsymbol{0} & \dots & \boldsymbol{0} & \boldsymbol{0} & \boldsymbol{0} \\ \boldsymbol{0} & \boldsymbol{I} & (\Delta x)^2 \tilde{\mathcal{L}}_1 - 2\boldsymbol{I} & \boldsymbol{I} & \dots & \boldsymbol{0} & \boldsymbol{0} & \boldsymbol{0} \\ \vdots & & & & \ddots & & & \vdots \\ \boldsymbol{0} & \boldsymbol{0} & \boldsymbol{0} & \boldsymbol{0} & \dots & \boldsymbol{0} & \boldsymbol{I} & (\Delta x)^2 \tilde{\mathcal{L}}_1 - 2\boldsymbol{I} \end{bmatrix}.$$
$$(31)$$

With this two-dimensional Laplacian matrix, the state vector $\boldsymbol{u}_h \in \mathbb{R}^{N^2}$ is advanced similarly via solving

$$\underbrace{\left( \boldsymbol{I} - \nu \Delta t \tilde{\mathcal{L}}_2 \right)}_{:=\boldsymbol{A} \in \mathbb{R}^{N^2 \times N^2}} \boldsymbol{u}_h^{[t+1]} = \boldsymbol{u}_h^{[t]}. \qquad (32)$$

Similar to the one-dimensional BTCS scheme, the system matrix is constant. The right-hand side assembly is again the identity.

### D.4 HEAT DIFFUSION IN 3D

The equation for heat diffusion in three dimensions reads

$$\frac{\partial u}{\partial t} = \nu \underbrace{\left( \frac{\partial^2 u}{\partial x^2} + \frac{\partial^2 u}{\partial y^2} + \frac{\partial^2 u}{\partial z^2} \right)}_{\Delta u}. \qquad (33)$$

We again use homogeneous Dirichlet boundary conditions on all six sides of the unit cube

$$u(t,0,y,z) = 0 = u(t,1,y,z), \quad u(t,x,0,z) = 0 = u(t,x,1,z), \quad u(t,x,y,0) = 0 = u(t,x,y,1). \qquad (34)$$

We consider the unit-cube $\Omega = (0,1)^3$ with $N$ degrees of freedom per dimension, in total $N^3$.

The Laplacian matrix in three dimensions can again be written in a block structure using the two-dimensional Laplacian $\tilde{\mathcal{L}}_2$ and appropriately sized identity matrices $\boldsymbol{I}$

$$\tilde{\mathcal{L}}_3 = \frac{1}{(\Delta x)^2} \begin{bmatrix} (\Delta x)^2 \tilde{\mathcal{L}}_2 - 2\boldsymbol{I} & \boldsymbol{I} & \boldsymbol{0} & \boldsymbol{0} & \dots & \boldsymbol{0} & \boldsymbol{0} & \boldsymbol{0} \\ \boldsymbol{I} & (\Delta x)^2 \tilde{\mathcal{L}}_2 - 2\boldsymbol{I} & \boldsymbol{I} & \boldsymbol{0} & \dots & \boldsymbol{0} & \boldsymbol{0} & \boldsymbol{0} \\ \boldsymbol{0} & \boldsymbol{I} & (\Delta x)^2 \tilde{\mathcal{L}}_2 - 2\boldsymbol{I} & \boldsymbol{I} & \dots & \boldsymbol{0} & \boldsymbol{0} & \boldsymbol{0} \\ \vdots & & & & \ddots & & & \vdots \\ \boldsymbol{0} & \boldsymbol{0} & \boldsymbol{0} & \boldsymbol{0} & \dots & \boldsymbol{0} & \boldsymbol{I} & (\Delta x)^2 \tilde{\mathcal{L}}_2 - 2\boldsymbol{I} \end{bmatrix}.$$
$$(35)$$

With this three-dimensional Laplacian matrix, the state vector $\boldsymbol{u}_h \in \mathbb{R}^{N^3}$ is advanced similarly via solving

$$\underbrace{\left( \boldsymbol{I} - \nu \Delta t \tilde{\mathcal{L}}_3 \right)}_{:=\boldsymbol{A} \in \mathbb{R}^{N^3 \times N^3}} \boldsymbol{u}_h^{[t+1]} = \boldsymbol{u}_h^{[t]}. \qquad (36)$$

Similar to the one-dimensional BTCS scheme, the system matrix is constant. The right-hand side assembly is again the identity.

### D.5 BURGERS IN 1D

The Burgers equation on the one-dimensional unit interval in non-conservative form with periodic boundary conditions reads

$$\frac{\partial u}{\partial t} + u\frac{\partial u}{\partial x} = \nu\frac{\partial^2 u}{\partial x^2} \qquad u(t,0) = u(t,1). \tag{37}$$

We will discretize the domain into $N+1$ grid points. Under periodic boundaries, only one of the boundary points can be eliminated. By convention, we choose the right-most point, leaving $N$ degrees of freedom (including the left-most point). The matrix associated with the discretized second derivative in one dimension now reads

$$\mathcal{L}_1 := \frac{1}{(\Delta x)^2}\begin{bmatrix} -2 & 1 & 0 & \dots & 0 & 1 \\ 1 & -2 & 1 & \dots & 0 & 0 \\ 0 & 1 & -2 & \dots & 0 & 0 \\ \vdots & & \ddots & & \vdots & \\ 1 & 0 & 0 & \dots & 1 & -2 \end{bmatrix}. \tag{38}$$

It differs from the former Laplacian matrix in that it is not exclusively tri-diagonal but also has entries in the top right and bottom left corners. Additionally, we now have $\Delta x = 1/N$.

The convection term requires special treatment because of its nonlinearity and the advection characteristic of the first derivative. Let $\mathcal{F}_1$ and $\mathcal{B}_1$ represent the forward or backward approximation of the first derivative in one dimension on periodic boundaries, respectively, via

$$\mathcal{F}_1 := \frac{1}{\Delta x}\begin{bmatrix} -1 & 1 & 0 & \dots & 0 & 0 \\ 0 & -1 & 1 & \dots & 0 & 0 \\ 0 & 0 & -1 & \dots & 0 & 0 \\ \vdots & & \ddots & & \vdots & \\ 1 & 0 & 0 & \dots & 0 & -1 \end{bmatrix} \qquad \mathcal{B}_1 := \frac{1}{\Delta x}\begin{bmatrix} 1 & 0 & 0 & \dots & 0 & 1 \\ -1 & 1 & 0 & \dots & 0 & 0 \\ 0 & -1 & 1 & \dots & 0 & 0 \\ \vdots & & \ddots & & \vdots & \\ 0 & 0 & 0 & \dots & -1 & 1 \end{bmatrix} \tag{39}$$

Again, note the element in the corner entries of the matrices. Then, we can build an upwind differentiation matrix based on the winds $\boldsymbol{w}_h$

$$\Gamma_1(\boldsymbol{w}_h) = \text{diag}\left(\underbrace{\max\left(\frac{\boldsymbol{s}_{-1}(\boldsymbol{w}_h) + \boldsymbol{w}_h}{2}, 0\right)}_{\text{positive winds}}\right)\mathcal{B}_1 + \text{diag}\left(\underbrace{\max\left(\frac{\boldsymbol{s}_1(\boldsymbol{w}_h) + \boldsymbol{w}_h}{2}, 0\right)}_{\text{negative winds}}\right)\mathcal{F}_1. \tag{40}$$

Deducing the positive and negative winds from neighboring averages (using the periodic forward shift $s_{-1}$ and backward shift $s_1$ operators) is necessary to have correct movement if the winds change sign over the domain. If we use the discrete state vector $\boldsymbol{u}_h$ as winds $\boldsymbol{w}_h$, we can discretize the continuous equation via the method of lines as

$$\frac{\mathrm{d}\boldsymbol{u}_h}{\mathrm{d}t} + \Gamma_1(\boldsymbol{u}_h)\boldsymbol{u}_h = \nu\mathcal{L}_1\boldsymbol{u}_h. \tag{41}$$

Naturally, the spatial discretization of a *nonlinear* PDE leads to a system of *nonlinear* ODEs. To fully resolve the nonlinearity, one could resort to a Newton-Raphson or a quasi-Newton method. However, for simplicity, we will apply the trick to linearize the upwind matrix using the state *previous in time* (Turek, 1999), which gives

$$\frac{\boldsymbol{u}_h^{[t+1]} - \boldsymbol{u}_h^{[t]}}{\Delta t} + \Gamma_1(\boldsymbol{u}_h^{[t]})\boldsymbol{u}_h^{[t+1]} = \nu\mathcal{L}_1\boldsymbol{u}_h^{[t+1]}. \tag{42}$$

This can be rearranged into the standard form as

$$\underbrace{\left(\boldsymbol{I} + \Delta t\Gamma_1(\boldsymbol{u}_h^{[t]}) - \Delta t\nu\mathcal{L}_1\right)}_{=\boldsymbol{A}=\Lambda(\boldsymbol{u}_h^{[t]})}\boldsymbol{u}_h^{[t+1]} = \boldsymbol{u}_h^{[t]}. \tag{43}$$

As such, the system matrix is dependent on the previous state in time, i.e., it is dependent on the previous variables in the compute graph. The right-hand side assembly routine is again the identity.

However, different from before is that the system matrix $A$ is now asymmetric, which necessitates special linear solvers.

Iteration over re-assembly and solution is possible but we omit this for simplicity, accepting that the *nonlinear* residuum is not fully converged. This introduces an error of order $\mathcal{O}(\Delta t)$, which is acceptable since our temporal discretization is first order. It must be noted that despite the nonlinear residuum is not fully converged, the linear residuum associated with the linearization will be, assuming we use the converged solver $\mathcal{P}$.

### D.6 BURGERS IN 2D

While we did not investigate any experiments with a two-dimensional Burgers solver, we will still present it here as it naturally helps with understanding the two-dimensional Navier-Stokes solver.

In two dimensions, the continuous solution function to the Burgers PDE becomes vector-valued with two channels. Using a symbolic notation, we can write

$$\frac{\partial \boldsymbol{u}}{\partial t} + (\boldsymbol{u} \cdot \nabla)\boldsymbol{u} = \nu \Delta \boldsymbol{u}. \tag{44}$$

An alternative way to write the two-dimensional Burgers equation is in its two components

$$
\begin{aligned}
\frac{\partial u_1}{\partial t} + u_1 \frac{\partial u_1}{\partial x} + u_2 \frac{\partial u_1}{\partial y} &= \nu \left( \frac{\partial^2 u_1}{\partial x^2} + \frac{\partial^2 u_1}{\partial y^2} \right), \\
\frac{\partial u_2}{\partial t} + u_1 \frac{\partial u_2}{\partial x} + u_2 \frac{\partial u_2}{\partial y} &= \nu \left( \frac{\partial^2 u_2}{\partial x^2} + \frac{\partial^2 u_2}{\partial y^2} \right).
\end{aligned}
\tag{45}
$$

Our domain $\Omega$ is again the unit square.

Under a two-dimensional biperiodic domain, both the right-most and the top-most boundary nodes are eliminated, see figure 12a. We consider the unit-square $\Omega = (0, 1)^2$ with equally many degrees of freedom per dimension, in total $N^2$.

The Laplacian matrix in two dimensions can be written in a block structure using the one-dimensional Laplacian $\tilde{\mathcal{L}}_1$ and appropriately sized identity matrices $\boldsymbol{I}$

$$
\mathcal{L}_2 = \frac{1}{(\Delta x)^2}
\begin{bmatrix}
(\Delta x)^2 \mathcal{L}_1 - 2\boldsymbol{I} & \boldsymbol{I} & \boldsymbol{0} & \boldsymbol{0} & \dots & \boldsymbol{0} & \boldsymbol{0} & \boldsymbol{I} \\
\boldsymbol{I} & (\Delta x)^2 \mathcal{L}_1 - 2\boldsymbol{I} & \boldsymbol{I} & \boldsymbol{0} & \dots & \boldsymbol{0} & \boldsymbol{0} & \boldsymbol{0} \\
\boldsymbol{0} & \boldsymbol{I} & (\Delta x)^2 \mathcal{L}_1 - 2\boldsymbol{I} & \boldsymbol{I} & \dots & \boldsymbol{0} & \boldsymbol{0} & \boldsymbol{0} \\
\vdots & & & & \ddots & & & \vdots \\
\boldsymbol{I} & \boldsymbol{0} & \boldsymbol{0} & \boldsymbol{0} & \dots & \boldsymbol{0} & \boldsymbol{I} & (\Delta x)^2 \mathcal{L}_1 - 2\boldsymbol{I}
\end{bmatrix}.
\tag{46}
$$

Note the identity blocks in the top right and bottom left corner.

Let $\mathcal{F}_{2,1}$ and $\mathcal{F}_{2,2}$ be the forward derivative operator in two dimensions in the direction of dimensions one and two, respectively. Moreover, $\mathcal{B}_{2,1}$ and $\mathcal{B}_{2,2}$ are the same for the backward derivative operator. Hence, we can build an upwinding operator for the first direction as

$$
\Gamma_{2,1}(\boldsymbol{w}_h) = \text{diag} \left( \underbrace{\max\left( \frac{\boldsymbol{s}_{-1}(\boldsymbol{w}_{h,1}) + \boldsymbol{w}_{h,1}}{2}, 0 \right)}_{\text{positive winds}} \right) \mathcal{B}_{2,1} + \text{diag} \left( \underbrace{\max\left( \frac{\boldsymbol{s}_1(\boldsymbol{w}_{h,1}) + \boldsymbol{w}_{h,1}}{2}, 0 \right)}_{\text{negative winds}} \right) \mathcal{F}_{2,1},
\tag{47}
$$

which only needs the winds in direction one. Similarly, we get for the other direction

$$
\Gamma_{2,2}(\boldsymbol{w}_h) = \text{diag} \left( \underbrace{\max\left( \frac{\boldsymbol{s}_{-1}(\boldsymbol{w}_{h,2}) + \boldsymbol{w}_{h,2}}{2}, 0 \right)}_{\text{positive winds}} \right) \mathcal{B}_{2,2} + \text{diag} \left( \underbrace{\max\left( \frac{\boldsymbol{s}_1(\boldsymbol{w}_{h,2}) + \boldsymbol{w}_{h,2}}{2}, 0 \right)}_{\text{negative winds}} \right) \mathcal{F}_{2,2}.
\tag{48}
$$

Let us combine these into one joint upwind discretization

$$\Gamma_2(\boldsymbol{w}_h) = \Gamma_{2,1}(\boldsymbol{w}_h) + \Gamma_{2,2}(\boldsymbol{w}_h) \tag{49}$$

This allows for discretizing the component-wise equation 45 first in space via the method of lines, afterward similarly in time as before to get the Oseen problem

$$\frac{\boldsymbol{u}_{h,1}^{[t+1]} - \boldsymbol{u}_{h,1}^{[t]}}{\Delta t} + \Gamma_2(\boldsymbol{u}_h^{[t]})\boldsymbol{u}_{h,1}^{[t+1]} = \nu\mathcal{L}_2\boldsymbol{u}_{h,2}^{[t+1]},$$
$$\frac{\boldsymbol{u}_{h,2}^{[t+1]} - \boldsymbol{u}_{h,2}^{[t]}}{\Delta t} + \Gamma_2(\boldsymbol{u}_h^{[t]})\boldsymbol{u}_{h,2}^{[t+1]} = \nu\mathcal{L}_2\boldsymbol{u}_{h,2}^{[t+1]}. \tag{50}$$

Note that for the discretization of each component, we need both components from the previous time level to assemble the upwinding matrix. We can re-arrange and write the system jointly as

$$\underbrace{\begin{bmatrix} \boldsymbol{I} + \Delta t\Gamma_2(\boldsymbol{u}_h^{[t]}) - \Delta t\nu\boldsymbol{B}_2 & \boldsymbol{0} \\ \boldsymbol{0} & \boldsymbol{I} + \Delta t\Gamma_2(\boldsymbol{u}_h^{[t]}) - \Delta t\nu\boldsymbol{B}_2 \end{bmatrix}}_{=\boldsymbol{A}=\Lambda(\boldsymbol{u}_h^{[t]})} \underbrace{\begin{bmatrix} \boldsymbol{u}_{h,1}^{[t+1]} \\ \boldsymbol{u}_{h,2}^{[t+1]} \end{bmatrix}}_{=\boldsymbol{u}_h^{[t+1]}} = \underbrace{\begin{bmatrix} \boldsymbol{u}_{h,1}^{[t]} \\ \boldsymbol{u}_{h,2}^{[t]} \end{bmatrix}}_{=\beta(\boldsymbol{u}_h^{[t]})}. \tag{51}$$

The assembly routine for the system matrix yields a block-diagonal structure while the right-hand side assembly is just the identity.

## D.7 COUPLED NAVIER-STOKES IN 2D

We solve the incompressible Navier-Stokes equations which can be written in symbolic notation as

$$\frac{\partial \boldsymbol{u}}{\partial t} + (\boldsymbol{u} \cdot \nabla)\boldsymbol{u} = -\nabla p + \nu\nabla^2\boldsymbol{u}, \tag{52}$$
$$\nabla \cdot \boldsymbol{u} = 0. \tag{53}$$

The domain is again unit-square with bi-periodic boundary conditions. This is a system of partial differential equations for three unknowns, which are the two components of the velocity $\boldsymbol{u}$ and the scalar pressure field $p$. The pressure acts as a constraint to enforce incompressibility given by the second equation. The coupling of velocity and pressure can be challenging, but a simple way for a finite difference discretization is the staggered grid (Harlow & Welch, 1965). As depicted in figure 12b, it also accounts for bi-periodicity by ignoring the top-most and right-most grid points but uses different locations to store the three unknowns. In this setting, there are equally many degrees of freedom per variable and direction. Each variable contributes $N^2$ entries. Hence, in total, there are $3N^2$.

Evaluating the convection term for both derivative directions requires mapping between the two staggered representations of the velocity grid. Let us denote the mapping operator $\mathcal{M}_1$ that moves all variables to the grid representation of the first velocity component. The operator $\mathcal{M}_2$ does the same for the second velocity component. These operators can be easily realized via bi-linear interpolation, which in the uniform cartesian grid simply amounts to the average of the four neighbors. Then the convection operator for velocity components one and two are $\Gamma_2(\mathcal{M}_1(\boldsymbol{u}_h))$ and $\Gamma_2(\mathcal{M}_2(\boldsymbol{u}_h))$, respectively.

The discretization of the pressure gradient and the velocity divergence requires a mapping between the velocity and the pressure representations. It turns out that a forward derivative $\mathcal{F}_{2,1}$ or $\mathcal{F}_{2,2}$ also maps from the velocity representation to the pressure representation. Hence, we can define the divergence operator on velocity components as $\boldsymbol{D}_1 = \mathcal{F}_{2,1}$ and $\boldsymbol{D}_2 = \mathcal{F}_{2,2}$, respectively. Vice versa, the gradient operators mapping from pressure to velocity representations are the backward differences, i.e., $\boldsymbol{G}_1 = \mathcal{B}_{2,1}$ and $\boldsymbol{G}_2 = \mathcal{B}_{2,2}$.

 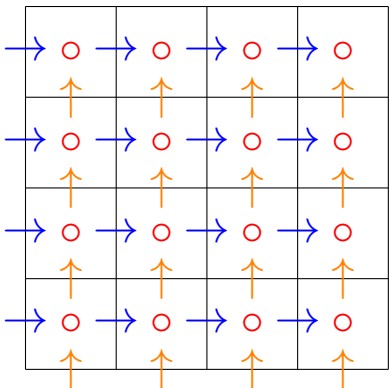

(a) Bi-periodic (collocated) grid. The top-most and right-most vertices have no degrees of freedom due to periodicity.

(b) Backward Staggering for bi-periodic domain. Horizontal arrows denote the location degrees of freedom for the x-velocity are saved. Vertical arrows correspond to y-velocity. Red circles are for pressure.

Figure 12: Arrangement of degrees of freedom on two-dimensional bi-periodic domains.

The method of line discretization with the linearization of convection matrices around the previous state in time then yields

$$\frac{\boldsymbol{u}_{h,1}^{[t+1]} - \boldsymbol{u}_{h,1}^{[t]}}{\Delta t} + \Gamma_2(\mathcal{M}_1(\boldsymbol{u}_h^{[t]}))\boldsymbol{u}_{h,1}^{[t+1]} = -\boldsymbol{G}_1\boldsymbol{u}_{h,3}^{[t+1]} + \nu\mathcal{L}_2\boldsymbol{u}_{h,1}^{[t+1]} \tag{54}$$

$$\frac{\boldsymbol{u}_{h,2}^{[t+1]} - \boldsymbol{u}_{h,2}^{[t]}}{\Delta t} + \Gamma_2(\mathcal{M}_2(\boldsymbol{u}_h^{[t]}))\boldsymbol{u}_{h,2}^{[t+1]} = -\boldsymbol{G}_2\boldsymbol{u}_{h,3}^{[t+1]} + \nu\mathcal{L}_2\boldsymbol{u}_{h,2}^{[t+1]} \tag{55}$$

$$\boldsymbol{D}_1\boldsymbol{u}_{h,1}^{[t+1]} + \boldsymbol{D}\boldsymbol{u}_{h,2}^{[t+1]} = \boldsymbol{0} \tag{56}$$

We can write this in matrix form as

$$\underbrace{\begin{bmatrix} \boldsymbol{I} + \Delta t\Gamma_2(\mathcal{M}_1(\boldsymbol{u}_h^{[t]})) - \Delta t\nu\mathcal{L}_2 & \boldsymbol{0} & \boldsymbol{G}_1 \\ \boldsymbol{0} & \boldsymbol{I} + \Delta t\Gamma_2(\mathcal{M}_2(\boldsymbol{u}_h^{[t]})) - \Delta t\nu\mathcal{L}_2 & \boldsymbol{G}_2 \\ \boldsymbol{D}_1 & \boldsymbol{D}_2 & \boldsymbol{0} \end{bmatrix}}_{:=\boldsymbol{A}=\Lambda(\boldsymbol{u}_h^{[t]})} \underbrace{\begin{bmatrix} \boldsymbol{u}_{h,1}^{[t+1]} \\ \boldsymbol{u}_{h,2}^{[t+1]} \\ \boldsymbol{u}_{h,3}^{[t+1]} \end{bmatrix}}_{\boldsymbol{u}_h^{[t+1]}} = \underbrace{\begin{bmatrix} \boldsymbol{u}_{h,1}^{[t]} \\ \boldsymbol{u}_{h,2}^{[t]} \\ \boldsymbol{0} \end{bmatrix}}_{=\beta(\boldsymbol{u}_h^{[t]})} \tag{57}$$

which reveals the saddle-point nature of the coupled Navier-Stokes system given by the zero block in the bottom right (Turek, 1999). Many popular solution techniques to the Navier-Stokes equations, like PISO (Issa et al., 1986), can be interpreted as efficient preconditioners to this coupled system as argued in Perot (1993). For simplicity and to not get nested iterative linear solvers, we solve the coupled system without further modifications, which is reasonable for the low employed resolution despite the considerably high condition number.

Similar to the Burgers simulators, the system matrix $\boldsymbol{A}$ is asymmetric and needs to be re-assembled from information given by the previous state in time. However, only the previous velocity components in time $\boldsymbol{u}_{h,1}^{[t]}$ and $\boldsymbol{u}_{h,2}^{[t]}$ are needed. The previous pressure state $\boldsymbol{u}_{h,3}^{[t]}$ does not affect the matrix assembly. It is also not relevant for the right-hand side assembly function $\beta(\cdot)$.

## E    MORE DETAILS ON PRDP PARAMETERS

**Stepping Threshold** $\tau_{\textbf{step}}$    The stepping criterion defines the threshold at which the validation metric is considered sufficiently flat before the physics is refined. This value should not exceed 1, as

values greater than 1 would trigger refinement only when the error increases. Higher values (less than 1) allow the metric to be flatter before the physics is refined. In other words, a lower value enables faster progressive refinement and trades off PR savings. Refinement can also be made faster by making larger steps in refinement $\Delta K$. In practice, we found that $\tau_{\text{step}}$ values between 0.9 and 1 with $\Delta K = 1$ sufficed in all our network training runs.

**Stopping Threshold $\tau_{\text{stop}}$** The stopping criterion on the checkpoint ratio governs how strictly the stagnation in network accuracy should be over different refinement levels. A lower value implies a stricter check, and progressive refinement would stop in a lower $K_{\text{max}}$ region, and vice versa. In other words, a lower value of $\tau_{\text{stop}}$ enables a more aggressive strategy for IC savings.

The grace window of intervals $\delta$ controls the length of lookback to calculate the stepping ratio $r$. This is helpful in the case of strong epoch-to-epoch oscillations. On the other hand, the effect of oscillations on the stepping ratio is mitigated through exponential smoothing. We found a window of 3 worked for most scenarios. The grace window goes hand-in-hand with the stepping criterion - a longer window requires stricter (smaller) $\tau_{\text{step}}$ values.

## F    EXPERIMENTAL SETUPS

An algorithmic description of Progressively Refine Differentiable Physics is given in algorithm 4. Our choice of PRDP parameters based on section E are listed in each problem setup as follows. Our scheduling is via changing the number of inner iterations via $\Delta K$ each time PRDP triggers a refinement. In figure 13, we demonstrate that this is identical to scheduling with relative tolerance thresholds.

**Validation Metric** For test cases based on the Heat diffusion, Burgers equation, and Navier Stokes, the validation metric we use for PRDP is the solution's error against a reference solution at a specific time step $t$, normed over space, and normalized against the reference solution. This error is mean-squared over the validation set. Mathematically, this can be written as

$$\text{Validation Error} = \frac{1}{|\mathcal{V}|} \sum_{i \in \mathcal{V}} \left( \frac{\|\boldsymbol{u}_i^{[t]} - \boldsymbol{u}_i^{[t],\text{r}}\|_2}{\|\boldsymbol{u}_i^{[t],\text{r}}\|_2} \right)^2 \tag{58}$$

where:

- $\boldsymbol{u}_i^{[t]}$: the predicted solution at time step $t$ for validation example $i$,
- $\boldsymbol{u}_i^{[t],\text{r}}$: the reference solution at time step $t$ for validation example $i$,
- $\|\cdot\|$: the norm over the spatial domain,
- $\mathcal{V}$: the validation set.

The corresponding specifics for the Poisson inverse problem are explained in F.1.

**Seed statistics** For the neural network training setups, we conducted trials with 10 different initialization seeds. Each run uses a different network initialization and a different stochastic minibatching, but the same data. The results shown in the work visualize the mean over these runs (shown as solid lines), along with its variability (represented by the shaded area). The shaded areas indicate the range within one standard deviation of the mean.

### F.1    POISSON EQUATION - INVERSE PROBLEM

The Poisson inverse problem is our simplest test example that incorporates differentiable physics into a learning pipeline for a doubly convex problem optimizing in a low-dimensional parameter space. We set up the discretized Poisson problem as described in section F.1 with degrees of freedom $N = 30$. The resulting linear system is parameterized, similar to section 2. The iterative linear solver applied to this system represents the physics $\mathcal{P}_K$ in this problem.

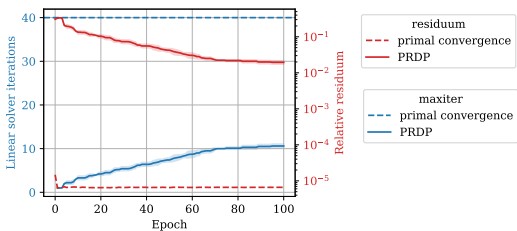

Figure 13: PRDP refines the physics by progressively increasing the solver iterations throughout training until training accuracy in terms of a validation metric stagnates. Progressively increasing the number of iterations is equivalent to progressively decreasing the tolerance threshold, as shown here for the example of the training the Navier-Stokes emulator of section 4.4,

---

**Algorithm 4:** Determine Whether to Refine Physics

---

**Input:** $\tau_{\text{step}} = 0.95$, $\tau_{\text{stop}} = 0.9$, $\delta = 3$, $\Delta e = 6$

**Data:** $\{L_{\text{val}}^{[e]}\}_e$

**Output:** Boolean indicating whether to refine

$\{\tilde{L}_{\text{val}}^{[e]}\}_e = \text{ema}(\{L_{\text{val}}^{[e]}\}_e, \Delta e)$        $\triangleright$ Smooth with Exponential Moving Average

$r = \tilde{L}_{\text{val}}^{[-1]} / \tilde{L}_{\text{val}}^{[-\delta]}$        $\triangleright$ Relative change to $\delta$ epochs prior

**if** $r > \tau_{step}$ **then**

     $r_c = \tilde{L}_{\text{val}}^{[-1]} / c$        $\triangleright$ Relative change to previous plateau

     **if** $r_c < \tau_{stop} \lor r_c > 1.0$ **then**

         $c \leftarrow \tilde{L}_{\text{val}}^{[-1]}$        $\triangleright$ Save current plateau

         **return** True        $\triangleright$ Refine!

     **else**

         **return** False        $\triangleright$ Do not further refine because we reached $K_{\max}$, $\implies$ yields IC savings

**else**

     **return** False        $\triangleright$ Do not refine to keep coarse physics for economic reasons, $\implies$ yields PR savings

---

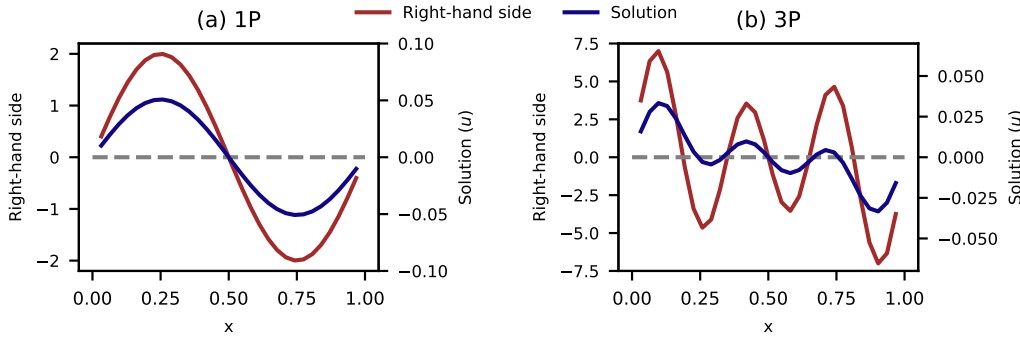

Figure 14: The Poisson PDE can be interpreted as the deformation of a thin string subject to a load. We consider two scenarios. In (a), the load (=right-hand side) only consists of one sine mode whose amplitude is scaled by a single (1P) $\theta$ parameter. The second case (b) uses the first three sine modes with a parameterized amplitude each.

For simplicity, we keep the system matrix $\boldsymbol{A}$ constant and only parameterize the right hand side $\boldsymbol{b}_h = \beta(\theta)$. The map $\beta$ is a sum of the first $P$ sine modes in the unit interval whose amplitudes are given by the parameter vector $\theta \in \mathbb{R}^P$.

$$\beta(\theta) = \sum_{i=1}^{P} \theta_i \sin(2i\pi x)$$

We design two inverse problems, one with a single sine mode ($P = 1$), and one with three sine modes ($P = 3$). A qualitative example is given in figure 14.

The outer optimization's objective is the MSE (mean-squared error) in the physics solution of $\boldsymbol{u}_h^{[K]}$ against a reference solution $\boldsymbol{u}_{h,\mathrm{r}}$. The reference is generated by direct solution of the linear system at a reference parameter value $\theta_{\mathrm{r}}$. In other words, $\boldsymbol{u}_{h,\mathrm{r}} = \boldsymbol{A}^{-1}\boldsymbol{b}_{h,r}$, where $\boldsymbol{b}_{h,r} = \beta(\theta_{\mathrm{r}})$. Optimization is performed using gradient descent algorithm.

**One-dimensional parameter space**  We use $\theta_{\mathrm{r}} = 2.0$, and an initial guess for gradient descent $\theta_{\mathrm{init}} = 5.0$. 170 update steps are performed with a constant learning rate of 275. For PRDP, we set the control parameter values to $\tau_{\mathrm{step}} = 0.92$, $\tau_{\mathrm{stop}} = 0.98$, $\delta = 2$. Training was started with $K_0 = 25$ linear solver iterations. At every refinement, it was incremented by $\Delta K = 10$. A relatively high value for $\Delta K$ and a aggressive refinement strategy with relatively smaller value of $\tau_{\mathrm{step}}$ are suitable for cases where $K_{\mathrm{max}}$ is significantly higher than the number of outer iterations. In this case, the physics converges at $K_\epsilon = 600$ (which is also $K_{\mathrm{max}}$ due to the double-convex problem). PRDP was successful at enabling 33% solver iteration savings

**Three-dimensional parameter space**  We use $\theta_{\mathrm{r}} = [0.62, 1.86, 5.1]$, and an initial guess for gradient descent $\theta_{\mathrm{init}} = [3.0, 3.0, 3.0]$. 750 update steps are performed when using Jacobi as the inner problem and 800 when using steepest descent, with a constant learning rate of 3500. The physics converged with 450 or 240 inner iterations when using Jacobi or steepest descent, respectively. In this case, a less aggressive PRDP strategy was suitable since the update steps were much higher than $K_\epsilon$. Hence, we found control parameter values $\tau_{\mathrm{step}} = 0.99$, $\tau_{\mathrm{stop}} = 0.99$, $\delta = 2$, with $\Delta K = 2$ and $K_0 = 1$ was suitable. PRDP was successful at enabling 40-50% savings for both Jacobi and steepest descent with both unrolled and implicit differentiation.

The results of these experiments are presented in figure 15.

### F.2  HEAT DIFFUSION - AUTOREGRESSIVE NEURAL EMULATOR TRAINING

In this setup, a neural emulator learns the diffusion equation solvers presented in sections D.2 and D.3. For the physics, we use a diffusivity of $\nu = 0.001$ and a time step size of $\Delta t = 1$, see equation 5. We discretize the space, into 32 grid points (including boundaries) in each dimension, which, after applying homogeneous Dirichlet boundary conditions, results in 30 and 900 degrees of freedom in the 1D and 2D cases, respectively. In the 3D case, we have 22 grid points including boundaries, i.e. 20 interior grid points, corresponding to 8000 degrees of freedom. The resulting linear system is solved using the Jacobi method. This constitutes the physics operator $\mathcal{P}_K$. We employ our own implementations of the Jacobi solver. The initial conditions are generated as a truncated Fourier series. For 1D, we sum the first 5 sine and cosine modes defined on the unit interval. For 2D, we use the products of the first 5 sine and cosine modes defined as:

$$u_0(x, y) = \sum_n (a_n \sin(2n\pi x)\sin(2n\pi y) + b_n \cos(2n\pi x)\cos(2n\pi y) +$$

$$c_n \sin(2n\pi x)\cos(2n\pi y) + d_n \cos(2n\pi x)\sin(2n\pi y))$$

We similarly extend this procedure to 3D. All amplitudes are randomly sampled from a uniform distribution $\mathcal{U}(-1, 1)$. A qualitative example of the dynamics is given in figure 16 for 1D, figure 17 for 2D, and figure 18 for 3D.

**Training Procedure**  We intentionally chose simple architectures as the focus of this work is not architecture but training methodology. For 1D and 2D cases, a multilayer perceptron (MLP) $\boldsymbol{f}_\theta$ is trained to emulate $\mathcal{P}_K$. For the 1D case, the MLP has 3 hidden layers, each of width 64, and for the

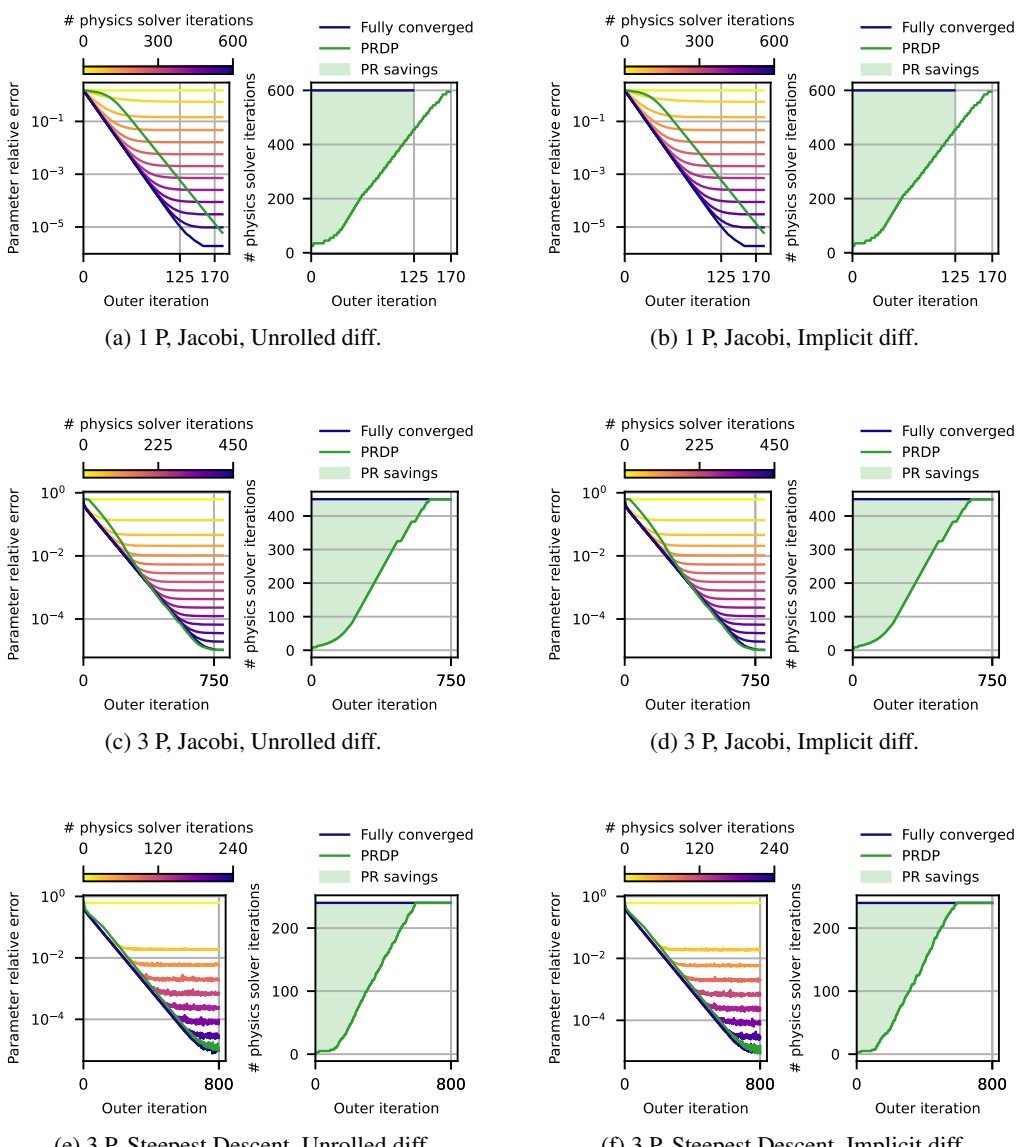

Figure 15: Results of the Poisson equation inverse problem visualizing the parameter error during training and the savings enabled by PRDP for both the 1 parameter and 3 parameter setup using different combinations of physics solvers (Jacobi, steepest descent) and solver differentiation methods (unrolled, implicit).

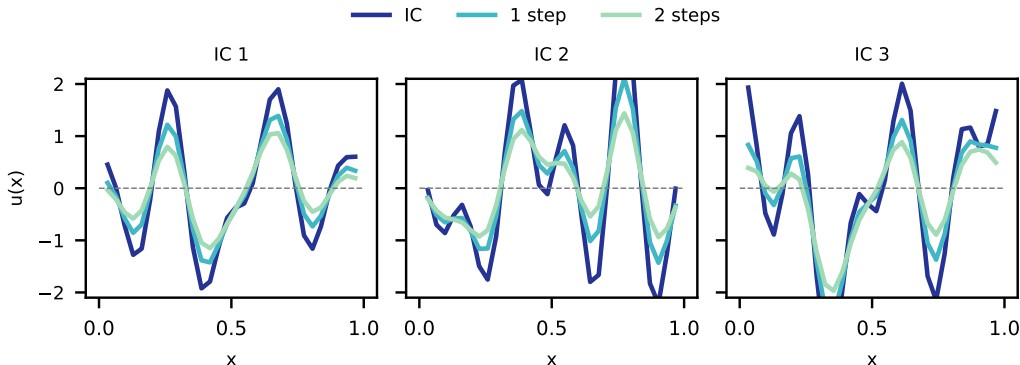

Figure 16: Example of three different trajectories over three temporal snapshots for the one-dimensional heat equation.

2D case, 3 hidden layers each of width of 3000. Corresponding to the physical problem's degrees of freedom, it has 30 input and 30 output nodes (900 for 2D). For the 3D case, we train a classic ResNet that supports 3 dimensional input and output tensors with 1 channel. It contains 6 blocks, each with 32 hidden channels. Its convolutions implement homogeneous Dirichlet boundary conditions. All networks are initialized with random weights. Non-linearization is enabled by applying ReLU activation after each hidden layer.

The compute graph consists of the neural emulator $f_\theta$ stepping from an initial condition to the first time step solution, and the physics operator $\mathcal{P}_K$ stepping from the neural network's solution to the solution at the second time-step. The nMSE between this solution at the second time-step against a reference solution forms the loss function for training. This *mixed-chain approach* (depicted in figure 22 allows gradients to flow through $\mathcal{P}_K$ and lets the network training to be informed with the physics's dynamics. We use implicit differentiation for the 1D case, and unrolled differentiation for the 2D and 3D cases. The custom VJP rules of the physics use the same iterative linear solvers as the primal solve and are resolved to the same number of iterations as the primal, as explained in section 2.1.

For the outer (training) problem, we use the Adam optimizer from the Optax library (DeepMind et al., 2020). The learning rate is scheduled as exponentially decaying with an initialization of $10^{-3}$, decay rate of 0.94 for 1D (0.9 for 2D), and 100 transition steps, while for 3D, the initialization is $10^{-4}$, with a decay rate of 0.92 and 100 transition steps. We train in mini-batches of 25 samples per iteration and for a total of 70 epochs in the 1D case, and 100 epochs in the 2D and 3D cases.

**Data Generation**  The reference solution at the second time-step is computed by two applications of the physics operator on the initial condition, where the iterative solver is replaced by a direct solver. Hence, the iterates $\{u^{[2][k]}\}_{k=0}^{K}$ converge to the reference solution. 205 samples are generated for training, with a train:validation split of 200:5.

**PRDP**  PRDP is controlled using the validation metric of equation 58 computed at $t = 2$, and refinement is applied to the iterative linear solver $\mathcal{P}_K$ that steps from $u^{[1]}$ to $u^{[2]}$. For the 1D case, we use parameter values $\tau_{\text{step}} = 0.98$, $\tau_{\text{stop}} = 0.9$, $\delta = 3$, exponential smoothing is performed over a time window of 8. The 2D case uses fewer epochs, hence a less aggressive refinement using $\tau_{\text{step}} = 0.9$ with a smoothing window of 6 was suitable. In the 3D case, which uses 100 epochs, used a less aggressive refinement with $\tau_{\text{step}} = 0.97$, similar to the 1D case, along with $\tau_{\text{stop}} = 0.9$, $\delta = 3$, exponential smoothing is performed over a time window of 6.

### F.3  BURGERS EQUATION - AUTOREGRESSIVE NEURAL EMULATOR

We solve the discretized Burgers equation presented in section D.5 on the unit interval with $\nu = 0.001$ and time step size $\Delta t = 0.01$. The spatial domain is discretized into 257 points in-

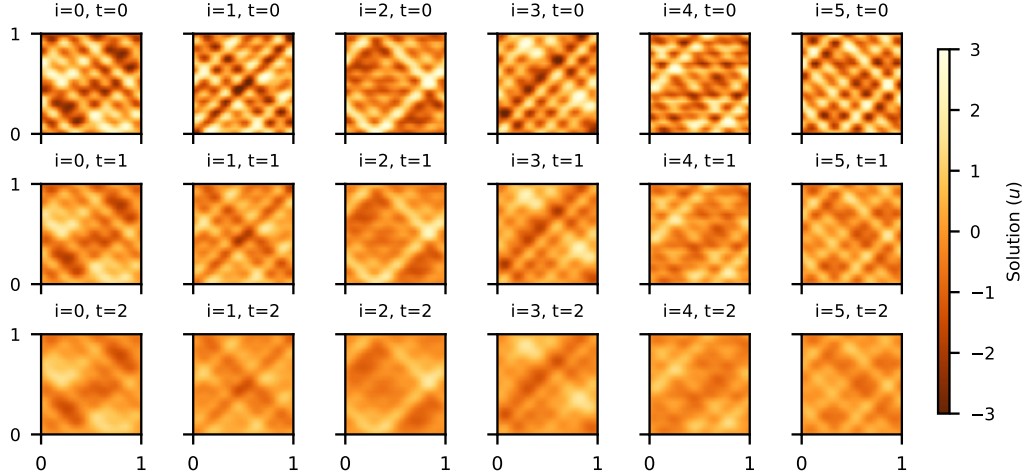

Figure 17: Example of six different trajectories over three temporal snapshots for the two-dimensional heat equation.

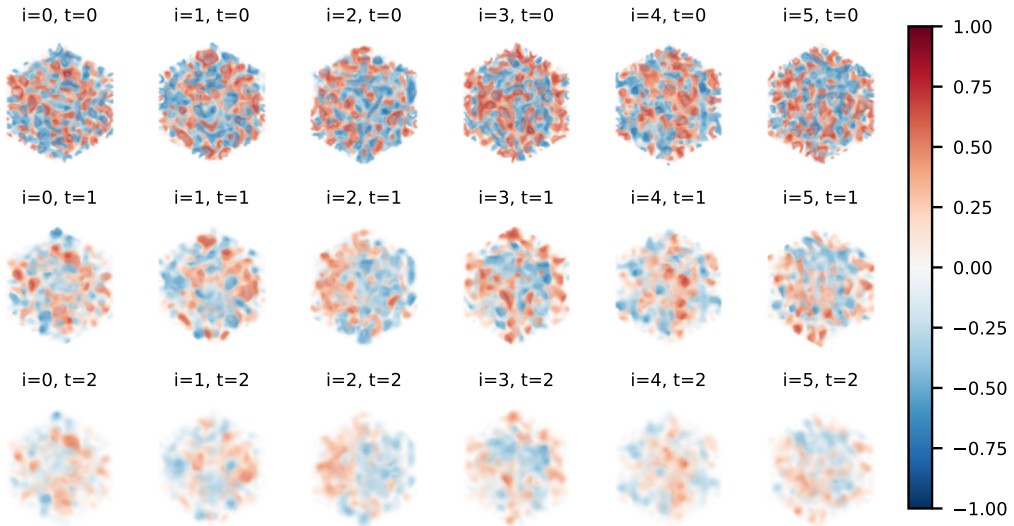

Figure 18: Example of six different trajectories over three temporal snapshots for the three-dimensional heat equation.

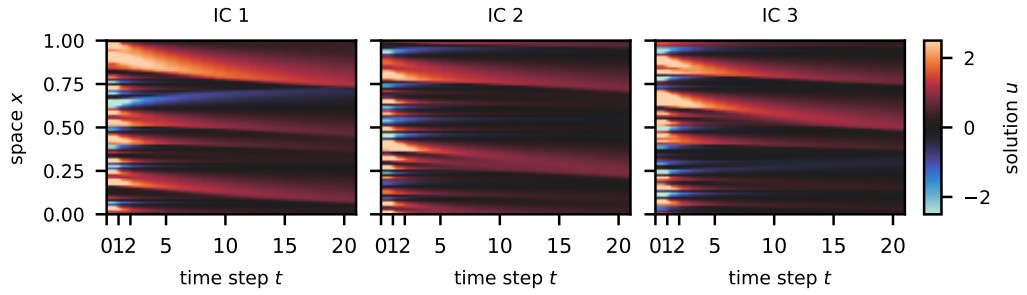

Figure 19: Three example trajectories of the one-dimensional Burgers equation.

cluding the boundaries, which results in $N = 256$ degrees of freedom when applying a periodic boundary condition. Since the system matrix $\boldsymbol{A}$ is now asymmetric, we use the GMRES linear solver (B.3) with restart set to 2. Although much higher restarts are possible and preferred in practice, we used fewer restarts to emphasize the difference between network accuracies trained using different number of GMRES iterations since our test problem was relatively easy to solve.

**Data Generation**  The initial conditions are generated as a truncated Fourier series as described in section F.2. The first 20 sine and cosine modes defined over the unit interval are summed, with their amplitudes sampled from a uniform distribution $\mathcal{U}(-1, 1)$. Qualitative example trajectories are given in figure 19

**Training Procedure**  We train a neural network $\boldsymbol{f}_\theta$ to learn the Burgers stepper which represents the physics $\mathcal{P}$. We use a classic ResNet architecture (He et al., 2016) consisting of 6 blocks, each with 32 hidden channels, and ReLU activation. The convolutions implement the physics' periodic boundary condition by circular padding. The weights are randomly initialized. We use our own implementation of the architecture using the Equinox library (Kidger & Garcia, 2021).

The loss function for training the network is defined similarly to the mixed-chain setup of section F.2. The ResNet steps from an initial condition to the first time step solution, and the physics $\mathcal{P}_K$ steps from the first to the second time-step solution. The nMSE error between this solution at the second time-step against a reference solution forms the loss function for training.

The reference solution at the second time-step is computed by two applications of the physics, with a direct solver replacing the iterative solver. 205 samples are generated for training, with a train:validation split of 200:5.

We use the Adam optimizer from the Optax library, with an exponential learning rate decay using an initialization of $10^{-3}$, a decay rate of 0.7, and 100 transition steps. We train in mini-batches of 25 samples per iteration.

**PRDP**  PRDP is controlled using the validation metric of equation 58 computed at $t = 5$, and refinement is applied to the iterative linear solver $\mathcal{P}_K$ that steps from $\boldsymbol{u}^{[1]}$ to $\boldsymbol{u}^{[2]}$. This presents an interesting application scenario, where the training objective uses a relatively short temporal rollout of 2 steps while PRDP favors generalization over a longer rollout at the fifth time step. We use PRDP parameter values $\tau_{\text{step}} = 0.9$, $\tau_{\text{stop}} = 0.9$, and $\delta = 3$. Due to high oscillations in the validation metric, we used exponential smoothing with a longer window of 15.

### F.4 NAVIER STOKES - NEURAL-HYBRID CORRECTOR LEARNING

In this example, we train a network to learn to correct for discretization errors in the solver described in section D.7. Such neural-hybrid methods apply to correction learning setups as investigated by Um et al. (2020). Our problem setup is visualized in figure 23.

We distinguish between a spatially-coarse solver, $\mathcal{P}_{s,K}$, and a spatially-fine solver, $\mathcal{P}_r$ (following the terminology of *source* and *reference* from Um et al. (2020)). In our test problem, $\mathcal{P}_{s,K}$ operates on a solution manifold that has half the spatial resolution of $\mathcal{P}_r$. We use 97 spatial degrees of freedom for $\mathcal{P}_r$ (i.e., a total of 291 degrees of freedom considering the three components of $\boldsymbol{u}$ ), and 48 for $\mathcal{P}_{s,K}$ (i.e., a total of 144). Both solvers are set up for $\nu = 0.0001$ and $\Delta t = 0.1$ and employ the GMRES solver (section B.3) with restart set to 8.

**Data generation**  To generate initial conditions for the data, we initialize a solution field in the manifold of $\mathcal{P}_r$ with each component sampled from the standard normal distribution. This random field is passed through a low pass filter, then normalized to have zero mean and standard deviation of 1, and finally passed through an incompressibility projection, giving $u_r^{[0]}$. Data trajectories $\{u_r^{[t]}\}_{t=0}^T$ are produced by repeated applications of $\mathcal{P}_r$ (uses a fully-converged GMRES) to $u_r^{[0]}$. These trajectories are then downsampled to the source manifold $\{u_s^{[t]}\}_{t=0}^T$. 205 trajectories are generated for training, with a train:validation split of 200:5. Qualitative example trajectories are given in figure 20.

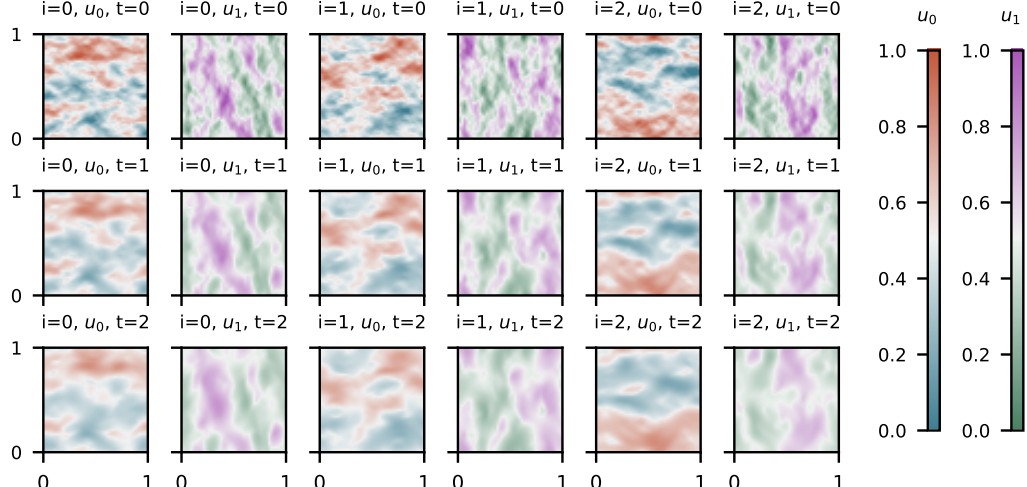

Figure 20: Three example trajectories of the Navier-Stokes scenario with the two velocity components each.

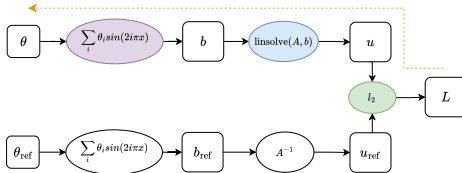

Figure 21: The compute graph and the reverse pass associated with the Poisson inverse problem.

**Training Procedure** The neural-hybrid model operates in the source manifold. We use a classic ResNet architecture (He et al., 2016) consisting of 3 blocks, each with 64 hidden channels. The convolutions implement the physics' periodic boundary condition by circular padding. The weights are randomly initialized. We implement the network architecture using the Equinox library (Kidger & Garcia, 2021).

The loss function is defined as the sum of MSE errors of the first and second time-step solutions. We use the Adam optimizer from the Optax library, with a cosine learning rate decay schedule using an initialization of $10^{-3}$ and decay_steps = 800. We train in mini batches of 25 samples per iteration and for a total of 100 epochs, i.e., 800 update steps.

**PRDP** PRDP is controlled using the validation metric of equation 58 computed on the x-velocity at $t = 5$. We used parameter values $\tau_{\text{step}} = 0.98$, $\tau_{\text{stop}} = 0.9$, and $\delta = 3$. For the exponential smoothing, we used a window size of 6. Since this problem setup has 2 executions of $\mathcal{P}_K$ (refer figure 23), three options arise for applying progressive refinement - a) progressively refining only the first execution, b) progressively refining only the second execution, and c) progressively refining both executions. We opted for c in order to gain savings from both solvers. In general, correction learning setups would have as many physics executions as the number of temporal unrolling steps in the loss function.

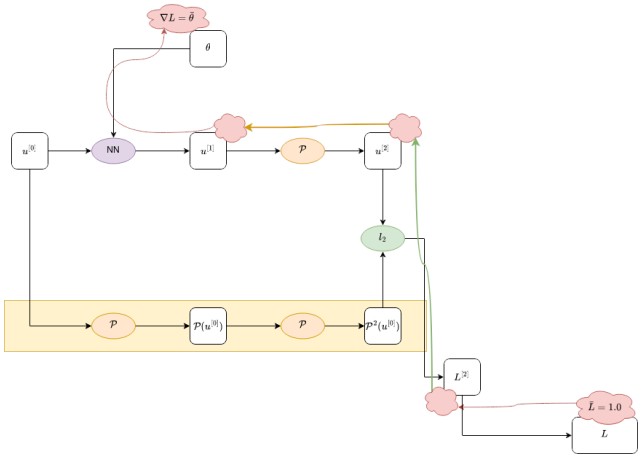

Figure 22: In the *mixed-chain approach* an emulator prediction is chained with an application of the differentiable solver, effectively to the training loss computed over a two step unrolling. Reference data is computed with full refinement, only the physics of the second step solver is refined with PRDP. The yellow reversely pointing error indicates the differentiable physics over the approximate solver.

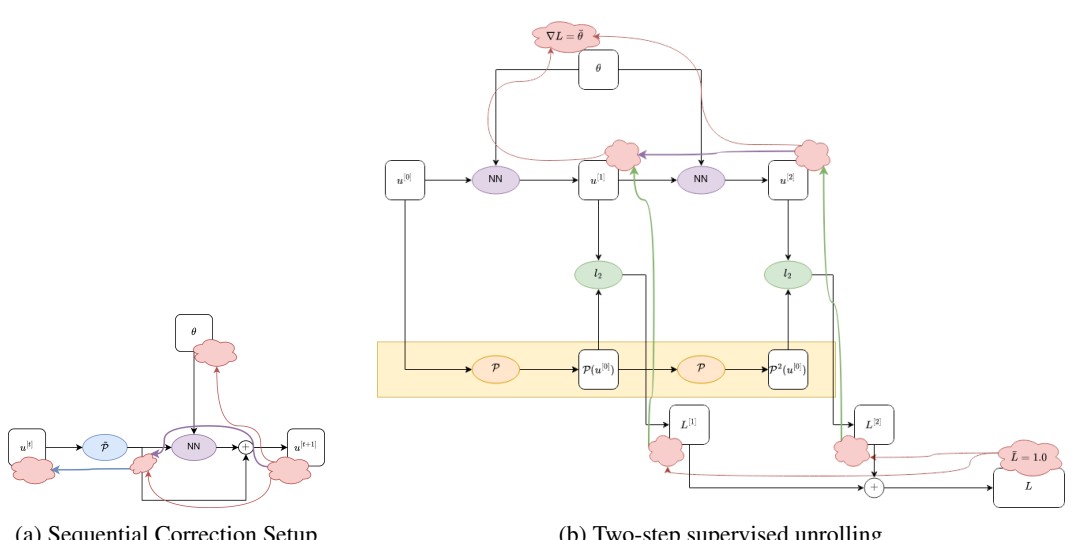

(a) Sequential Correction Setup          (b) Two-step supervised unrolling

Figure 23: The neural hybrid emulator is at the place of "NN" in the right figure. Each backpropagation-through-time also triggers the differentiation over the physics. Note that even if there is no differentiation over the first neural-hybrid emulator application (i.e., no differentiation over the first coarse solver usage), it is also affected by PRDP because we also limit its exactness in the primal execution. Here, one could think of it as "Progressively Refined Physics" instead of "Progressively Refined *Differentiable* Physics".

## G SUPPLEMENTARY EXPERIMENTAL RESULTS

### G.1 PRDP SAVINGS IN TERMS OF COMPUTE TIME

In figure 24, we present wall-clock training times with PRDP and with a fully converged differentiable solver. For the smaller 1D problems with small system matrices of the order of $\approx 100 \times 100$, other overhead in the compute graph dominates the training runtime. As such, the reduction in inner iterations does not translate to savings in wall clock time. Starting with 2D problems leading to larger matrices, decreases in training time converge to the reductions in inner iterations. For example, in the 3D heat emulator scenario, 81% reduction in inner iteration count aligned with wall clock time savings of 78%. As the share of iterative linear solves in the total training cost increases, e.g. in high-dimensional problems, so do the savings enabled by PRDP.

### G.2 NEURAL NETWORK PERFORMANCE - PRDP VS. BASELINE

The PRDP method is aimed at reducing the end-to-end computational cost of training neural networks with physics solvers in the loop, without significantly affecting the trained neural network's performance. This is depicted qualitatively in figure 6. In table 1, we provide the mean validation error values at the last epoch of training along with standard deviation over 10 seeds used for network initialization and stochastic minibatching.

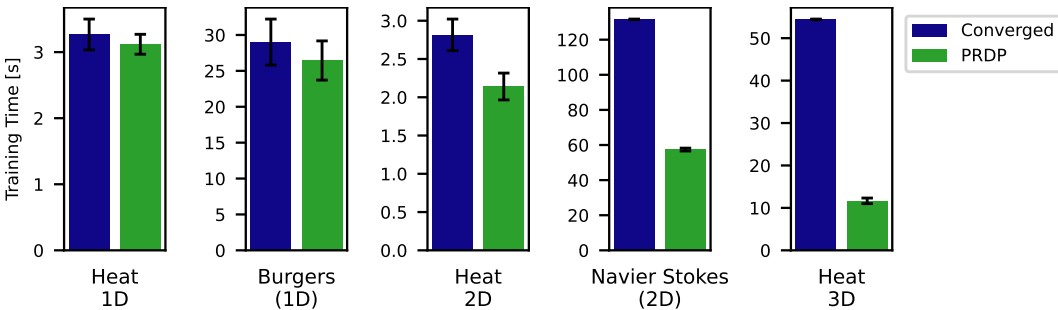

Figure 24: While all experiments reduced the number of inner iterations across the outer optimization (see section 4 bar charts), savings in training time become noticeable the larger and more complex the setup gets. Once the problem is of sufficient size, linear solve times dominate the training compute graph, and hence, PRDP can deliver the highest amount of savings in wall clock time. Eventually, savings in training compute time converge against the reductions in inner iterations.

Table 1: Numeric values of the validation accuracy of the neural emulators and neural-hybrid emulators according to the experiments of section 4. The mean and standard deviation are computed based on ten different random seeds that affected neural network initialization and stochastic minibatching.

| Outer training task | Inner physics | Val. error with Converged Physics $(\times 10^{-2})$ | Val. error with PRDP $(\times 10^{-2})$ |
|---|---|---|---|
| Neural Emulator | Heat 1D | $2.1 \pm 0.39$ | $2.1 \pm 0.38$ |
| Neural Emulator | Heat 2D | $1.4 \pm 0.08$ | $1.1 \pm 0.06$ |
| Neural Emulator | Heat 3D | $0.4 \pm 0.09$ | $0.5 \pm 0.10$ |
| Neural Emulator | Burgers | $5.5 \pm 1.38$ | $6.2 \pm 2.13$ |
| Neural-Hybrid Emulator | Navier Stokes | $0.6 \pm 0.05$ | $0.7 \pm 0.04$ |

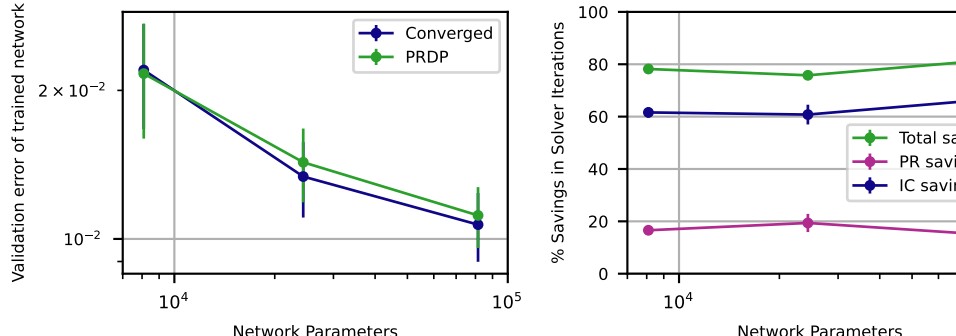

Figure 25: As expected with neural networks, the validation error falls over the number of parameters due to a larger network indicating a higher expressiveness. However, PRDP savings remain unaffected.

### G.3 PRDP SAVINGS VS. NEURAL NETWORK EXPRESSIVENESS

While we could not identify a single cause for the IC savings observed when solving the outer problem of learning in the high-dimensional, nonconvex weight space of a neural network, we hypothesized several contributing factors in the last paragraph of section 2.3. One hypothesis is that the inherently approximative nature of machine learning models plays a role. Specifically, limited model expressiveness—such as insufficient parameterization—might contribute to IC savings. In this view, a more expressive model would result in reduced IC savings.

To test this hypothesis, we conducted an ablation study by increasing the parameter count of the MLP emulator for the 1D Heat example in section 4.2 by an order of magnitude and retraining with PRDP. As expected, the larger model achieved higher accuracy, with a convergence rate of approximately $n^{-1/5}$. However, the impact on IC savings was minimal. Interestingly, the largest model exhibited slightly higher savings, likely due to triggering one fewer refinement level during training. Nonetheless, PRDP consistently yields savings of around 80% in terms of solver iterations across the varying network sizes.

### G.4 PRDP BASED ON A VALIDATION METRIC VS. ON TRAINING LOSS

We motivated Progressively Refined Differentiable Physics (PRDP) by showing how different physics refinement levels $K$ correspond to distinct plateaus in performance metrics other than training loss. This phenomenon is evident in figure 3 (a) and figure 4 (a). The PRDP control algorithm leverages this behavior by economically increasing the refinement levels, as visibly pronounced in figure 6 (a) and (b).

As introduced in section 3, the PRDP control algorithm relies on a performance metric that triggers refinement when it plateaus. In the main text, we used validation error for this purpose, as it exhibited clear plateaus for $K < K_{\max}$, and since hyperparameter tuning is typically done using a validation set. In this section, we ablate this choice by instead using the training loss as the guiding metric for PRDP. The results, shown in figure 26, reveal that training loss can also serve as a valid performance metric for PRDP.

Interestingly, when using training loss, PRDP adopts a slower refinement strategy since training loss often continues to decrease even after the validation loss has plateaued. While this approach yields slightly higher PR savings, the overall IC savings and final accuracy remain comparable.

Despite these findings, we recommend caution when applying PRDP with training loss as the guiding metric. For more complex problems, such as those with multi-modality or spurious minima, the training loss may be less reliable than validation loss. Additionally, as shown in figure 26, PRDP guided by validation loss results in smoother convergence for both training and validation losses. This smoother progression is likely a consequence of the more uniform refinement schedule, making validation loss the preferable choice for broader applications.

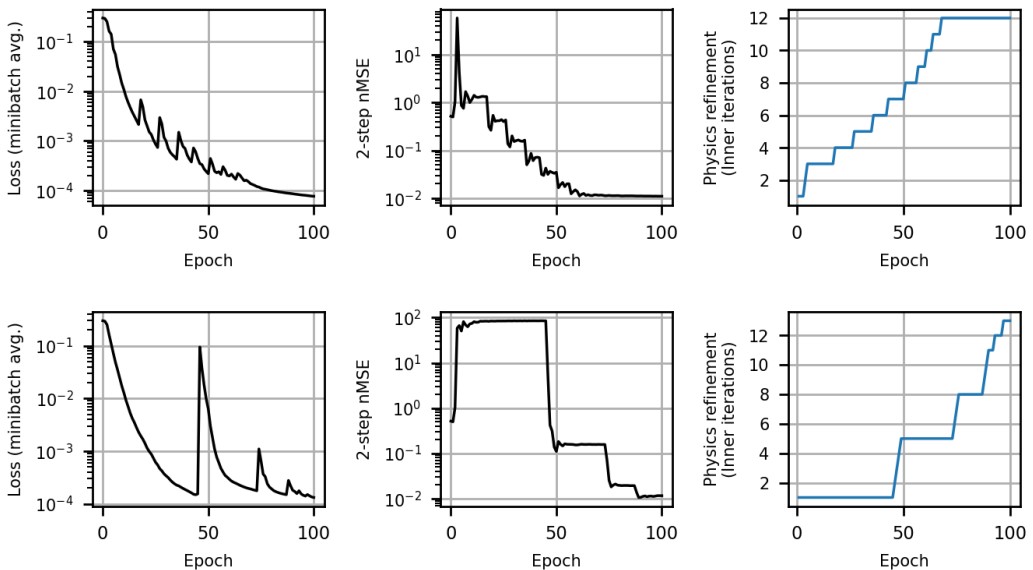

(a) Neural emulator learning for 2D heat equation: PRDP based on validation error (top) vs. training loss (bottom)

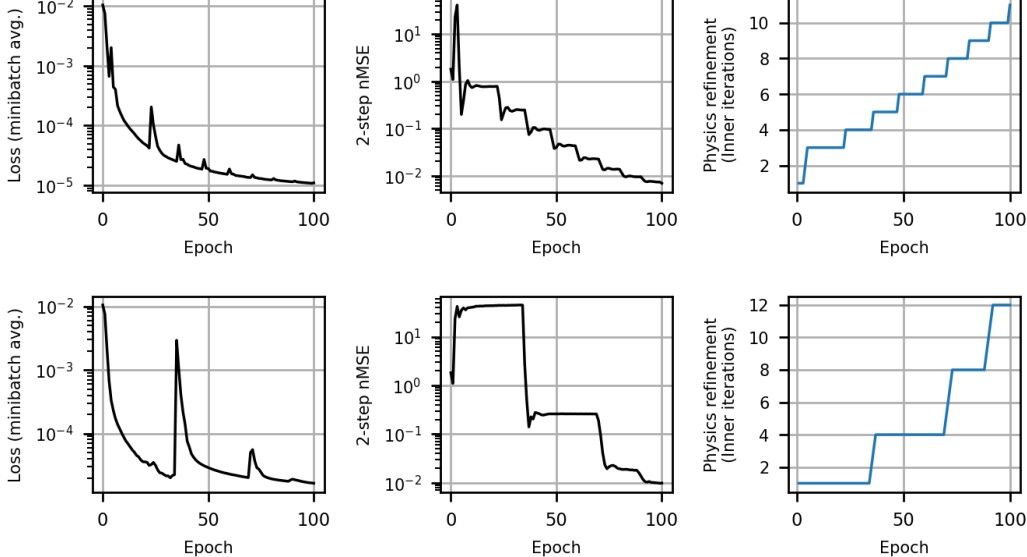

(b) Neural emulator learning for 3D heat equation: PRDP based on validation error (top) vs. training loss (bottom)

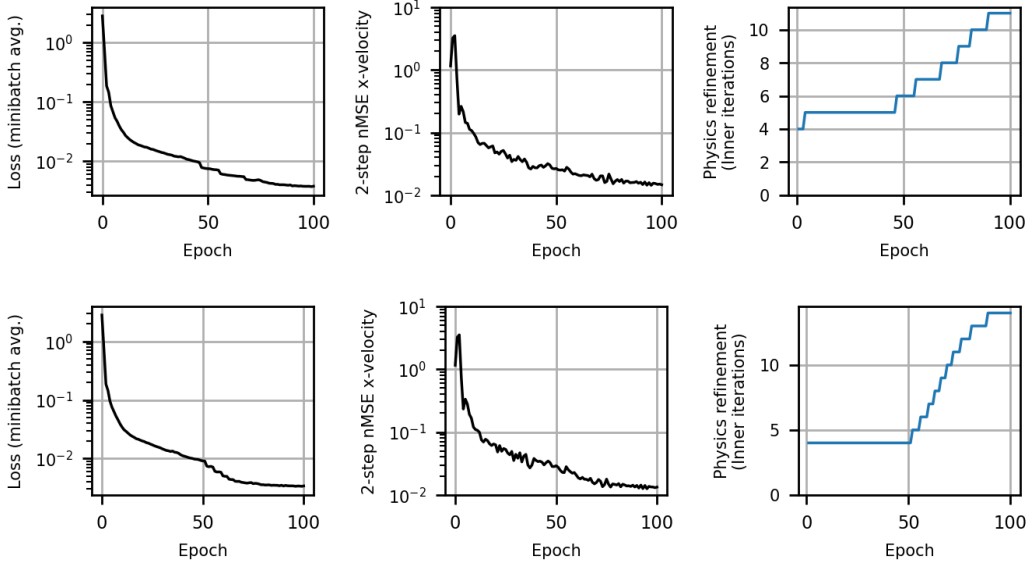

(c) Neural emulator learning for Burgers equation: PRDP based on validation error (top) vs. training loss (bottom)

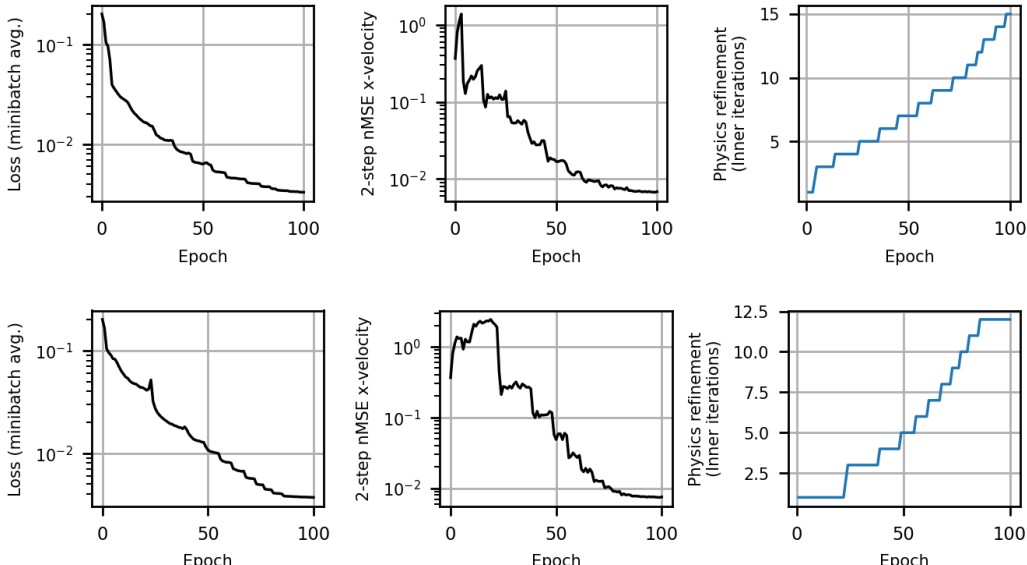

(d) Neural emulator learning for Navier Stokes equation: PRDP based on validation error (top) vs. training loss (bottom)

Figure 26: PRDP based on training loss also successfully trains the network but leads to a slower refinement strategy since the training loss oftentimes continues to reduce even if the validation metric plateaued.

