# OpenReview forum: "PRDP: Progressively Refined Differentiable Physics"
_ICLR.cc/2025/Conference — ICLR 2025 Poster_

### Official Review · Reviewer_h4zx · 2024-10-30

**Soundness:** 3
**Presentation:** 3
**Contribution:** 3
**Rating:** 6
**Confidence:** 3

**Summary:**

PRDP proposes an algorithm to reduce computational cost of training neural networks when a nested optimization problem is required during training. The authors first exhibit source of savings (IC and PR), before introducing their method. Specifically, this method consists in considering partially solving the inner problem in for the firsts steps of training and then refining this inner solution to more precise one as the training goes through. Moreover, the method considers not fully solving linear systems, since it may not greatly improve the performances. Finally, they evaluate their method on several PDE problems and instances.

**Strengths:**

- The paper is very complete.
- Several experiments are proposed and isolate and illustrate well the 2 kinds of cost reduction proposed.
- The proposed algorithm is easy to understand and well described.
- A lot of details about training and the physical systems studied are proposed in the appendices.
- The authors discuss impact and limitations of their work

**Weaknesses:**

- The paper is very technical and I think pseudo code or a scheme would help in the comprehension of the method. For example, it is unclear to me where is the neural network used in the global framework/experiments, what are inner/outer optimization problems in the examples... Despite understanding the overall idea and performances of the proposed algorithm, I think more explicit and easy-to-understand notations would help the comprehension. A more detailed example would help for a precise comprehension of the framework proposed (see questions section).

**Questions:**

-	Is this method only applicable in the context of Physical systems? It seems to me that this method could be more general and this being used in a broader range of applications, as soon as an iterative process incurs in the forward pass?
-	Could the author also provide a comparison of the differences of performances w/ and w/o PRDP? (see for example Fig 1 where it looks like there is a very little difference in performances, thus making me wondering how much is this loss)
-	Why is the algorithm based on validation losses? In what does these losses consist in? At inference, these validation values are not available?
-	What are the applications at inference? Once trained, what would be some application of NNs? Could this method be applied on new physical systems/PDEs/boundary or initial conditions/discretization?
-	In section 2, what does the subscript h stands for? Are the parameters $\theta$ the neural network parameters to be optimized through training?
-	On page 2, last paragraph it is stated that experiments are conducted on real world application, is the Navier-Stokes this example? These are synthetic data; in which sense do you consider this example as “real-world application”?
-	In example 2.2, are the $\theta$ parameters optimized with the outer step? This means that one wants to optimize the forcing terms? The application would be to find the forcing term associated to a recorded and given trajectory?
-	For the IC savings, I was wondering if the authors have tried to experiment if without NN, the performances would be better? My guess is that the introduction of a NN in the framework prevent the performances from being optimal, thus allowing for IC savings. What if the neural network size increases/its expressiveness improves? Are the performances better?
-	The main claim of the paper is computation savings. Could the author provide training times for their experiments? And a comparison of this solving time with standard methods?
-	What happen on cases where the losses plateau then decreases again? This situation could arise in some training of neural networks especially, when using a learning rate scheduler. How would behave the method in this context?
-	In section 4.4 why don’t you compare your results (performances, training times) with the method from Um et al.? Since the setting is the same and your method is supposed to improve training times, it would be interesting to evaluate the benefit with other methods existing. Moreover, in the related work several methods are cited, and could be used as a comparison.

---

> ### Author Response · Authors · 2024-11-21
>
> Dear reviewers,
>
> Thank you for your valuable inputs. We focussed our efforts towards clarity and accessibility, and we are glad for your feedback suggesting opportunities for better presentation. Since our work combines ideas from two slightly disconnected domains, i.e., differentiable physics and bilevel optimization, we acknowledge that certain domain-specific terminology and notation may not be naturally evident. Below we address specific concerns raised in your review.
>
> 1. **Pseudo Code:** Under Algorithm 4 in the appendix, we have provided a pseudo code for the PRDP control algorithm from section 3\. For better comprehension,
>    1. We will create a schematic that explains the basic idea of the algorithm more intuitively.
>    2. We will add pseudo-code of a typical solver-in-the-loop training pipeline in the main text that additionally shows where we invoke the PRDP control algorithm and the physics refinement.
> 2. **Where is the neural network used in the global framework/experiments:** The global framework is a neural network training pipeline where the gradients pass through an iterative physics solver. Figures 17-19 in the Appendix depict this framework, and Appendix E provides a detailed explanation on how the networks are trained, and how they are employed for inference.
> 3. **What are inner/outer optimization problems in the examples:** In all cases, the outer problem is an optimization problem that trains a neural network (or solves an inverse problem in the Poisson case), while the inner problem is the solution to the linear system that represents the physics (ref. Equation 1). These details are available in the appendix E. We value your feedback and will make suitable edits in the main text to present these details more explicitly.
> 4. **Notations:** We will add a paragraph summarising all notations in the appendix.
> 5. **Applications to other iterative processes:** Indeed, the core mechanisms of PRDP are not bound to just iterative linear solvers. We also argue that our approach is inspired by other fields of bi-level optimization as we discuss in the second paragraph of Section 5\. To the best of our knowledge, PRDP is the first time such a scheduling approach is used for contexts of training neural networks with differentiable numerical solvers. Conversely, we are optimistic that the aspects of PRDP will find their way back to the fields of hyperparameter optimization, meta learning, etc. This is especially noteworthy since our core motivation of Pedregosa (2016) works with machine learning models requiring a convex optimization fit. Solving linear systems can also be seen as a (quadratic) convex optimization albeit with sparse and structured system matrices (in the case of discretized PDE models) instead of dense data matrices.
> 6. **Performance difference when trained w/ and w/o PRDP:** With PRDP, we reduce training costs without significantly affecting performance. Indeed, there is a very small difference visible in Figure 1\. Conversely, we also see a very small improvement in performance (see e.g. figure 4 (b)). These minor differences were usually within bounds of the variance over random seeds. We will share the exact numbers in the revised version. In general, our results show that PRDP accelerates training while retaining the full accuracy at inference time.
> 7. **Why based on Validation Loss:** Thank you for this interesting question. Our choice of a validation metric is based on the following observations.
>    1. Previous approaches, i.e., Pedregosa (2016), implement progressive refinement through a sequence of tolerances. While this approach provides PR savings, it does not enable IC savings in problems where a certain level of incomplete inner refinement is sufficient for the network’s performance. In our approach, this refinement level is effectively identified by continuously examining a performance metric.
>    2. At first guess, one may pick training loss to serve as this performance metric. However, the training loss can be a misleading indicator. We observed in some experiments (e.g. emulator training for the heat equation) that with incompletely converged physics, the network training loss reduces while the validation error plateaued. Consequently, basing PRDP on training loss could result in a network that has low training loss but does not generalise to unseen data. By using validation errors, we make PRDP robust towards sufficiently refining the physics ensuring network performance.

---

> ### Author Response · Authors · 2024-11-21
>
> 8. **Validation values at inference:** During inference, PRDP does not play any role. It applies only to training the neural network. In problems where the inference involves not only the neural network but also the physics (e.g. correction learning setups like our Navier Stokes example), the physics is fully converged during inference.
> 9. **Applications of these NNs:** The neural emulators are trained to become cheap surrogates for forecasting problems modelled by PDEs. Please cf. our first reply to reviewer 1fLe. We acknowledge that the introduction can be enhanced by providing more details on the bigger picture.
> 10. **Notation:**
>     1. The subscript $h$ is taken from standard CFD textbooks to express the discretization of a continuous variable, where $h$ stands for the width of a cell in discretized space. Thank you for pointing this out; we will mention this explicitly in the main text.
>     2. Yes, $\\theta$ refers to the arguments for the outer optimization problem, i.e., the neural network parameters. The linear system of the physics follows the neural network’s computation (ref. compute graphs in figures 17-19), hence the linear system is indirectly parameterized by $\\theta$.
>
>     We will add a figure in the main text that provides a general overview of our experimental framework, clarifying the interaction of the neural networks and the physics.
>
> 11. **Navier Stokes \= real world?:** We used “real-world” synonymous with the difficulty the Navier-Stokes equation usually poses for numerical integration. It is a nonlinear system of equations, with an asymmetric advection characteristic and has a saddle-point structure. While our data is indeed synthetic, the step from our model to engineering CFD simulations (which are used to simulate the real world) is smaller than the step from the illustrative heat emulation to the Navier-Stokes. Hence, the Navier-Stokes example is the hardest test case of our submission.
> 12. **Section 2.2 problem setup:**
>     1. Yes, it is an inverse problem. We know the response displacement to the Poisson equation, then we model a forcing function with a free parameter and fit this parameter via comparing the predicted Poisson solution with our reference.
>     2. No, there is no trajectory because the Poisson equation is steady-state. Hence, the application is finding the forcing term associated with a given steady-state displacement.
> 13. **Performance of the test models:** Thank you for this question regarding the bigger picture. For our work, we were purposefully interested in running experiments with neural networks. Our work targets neural emulators/surrogates or neural operators that are trained through differentiable physics. This method has shown success in smaller fluid problems (Kochkov et al. and Um et al.) and most recently in weather and climate (NeuralGCM). In other words, performant models enabled by differentiable physics pipelines are a proven strategy. Despite the promise of neural-hybrid models, they often lack adoption since executing and differentiating over classical numerical solvers during training is costly. The focus of our work is not on improving the trained models, but rather to improve the training methodology.
> 14. **What if the neural network size increases/its expressiveness improves? Are the performances better?:** Based on your feedback, we have conducted a scaling test with the neural network parameter size for the heat 1d case. For an order of magnitude increase in the neural network size, the network’s accuracy improved by nearly a quarter of an order of magnitude, while the savings achieved by PRDP was nearly the same (79% and 81%). We will add the details to the revised pdf.
> 15. **Training times:**
>     1. We provide training times for the challenging Navier Stokes experiment in Figure 1 with a notable 62% improvement.
>     2. **Solving time:** As we pointed out in the previous points, our methodology pertains only to training but not inference.

---

> > ### Author Response · Authors · 2024-11-21
> >
> > 16. **Loss plateaus and decreases again (LR schedulers):** Thank you for this interesting case. PRDP handles this case through the algorithmic steps we describe below.
> >     1. When the validation metric is plateaued over epochs, PRDP checks whether it is also plateaued against a previous refinement level.
> >        1. If not, a refinement is invoked.
> >        2. Otherwise, no refinement is invoked.
> >
> >        If we understand correctly, your question refers to the second case (a.ii.).
> >
> >     2. At such a plateau, if the validation metric decreases again (e.g. due to learning rate annealing), PRDP’s checkpointing mechanism will record this decrease.
> >     3. At a subsequent loss plateauing, the checkpoint ratio $r\_c$ will indicate the earlier decrease and invoke a physics refinement.
> >
> > 	Hence, if a plateauing is caused by learning rate rather than physics refinement, and subsequently the loss decreases again, PRDP will successfully continue a judicious refinement of the physics.
> >
> > 17. **Comparison with Um et. al. and with other methods:** We designed the final Navier-Stokes experiment to represent Um et al. However, Um et al. deviates slightly: it uses an operator splitting approach (with semi-Lagrangian advection) to the NS equations, has different setups (like vortex shedding), and does more than two unrolled steps. On the other hand, our scenario uses a coupled solver and investigates decaying turbulence.
> >     We designed our Navier-Stokes experiment to be broadly inspired by the setup in Um et al., but there are significant methodological differences. While Um et al. employs an operator splitting approach (with semi-Lagrangian advection), investigates setups like vortex shedding, and uses more than two unrolled steps, our experiment instead focuses on a coupled solver approach to study decaying turbulence. These differences align with our emphasis on coupling solver fidelity with differentiable physics training pipelines.
> >     Regarding the other cited methods (Fung et al., 2021; Geng et al., 2021; Lorraine et al., 2020; Shaban et al., 2019; Bolte et al., 2023), their focus lies in adjusting the adjoint (in)accuracy, typically through static truncation of reverse-pass iterations, as seen in Equation 2\. While effective in their respective domains, these methods:
> >     1. Always execute a full primal pass, as they do not leverage primal (in)accuracy like PRDP, thereby achieving IC savings only in the reverse pass (loosely speaking, contributing only half the IC savings with PRDP).
> >     2. Are oftentimes static (i.e., do not adapt the truncated steps over the outer optimization)
> >    PRDP, by contrast, introduces savings in both the primal and reverse passes via its combined progressive refinement (PR) and incomplete convergence (IC) mechanisms. Thus, in scenarios involving sparse, structured linear systems arising from discretized PDE models—our primary focus—PRDP's achievable savings are likely the upper bound for savings strategies.
> >
> >      It’s important to note, however, that the methods referenced were not designed for differentiable physics pipelines. Instead, they target deep equilibrium models, hyperparameter optimization, or related machine learning contexts. These settings typically involve dense system matrices in their (implicit) reverse pass and nonconvex optimization tasks such as finding high-dimensional roots or fitting neural networks.
> >      Differentiable physics, as emphasised in our introduction and Section 2.1, represents a unique use case. Its linear solves in both primal and reverse passes allow PRDP to exploit structured sparsity and iterative solver dynamics more effectively. We show that this can be efficiently done across a wide range of scenarios including well-behaved symmetric positive definite linear matrices, asymmetric upwinding matrices, parameter-dependent matrices and saddle-point problems. These arose from PDE problems in 1D and 2D. Based on feedback by reviewer Skvr, we also added a 3D example, in which PRDP works equally well.

---

> > > ### Comment · Reviewer_h4zx · 2024-11-23
> > >
> > > I thank the author for their answers to my (numerous) questions.
> > > I think most of your answer should appear in the main part of your paper (e.g. 1, 2, 3) to improve the clarity of your paper and so that it is more straightforward for the reader to understand your method.
> > > 6. If your method is targeting to reduce the training time of neural networks, then I think including the training duration is required to illustrate the performances of your method.
> > > 7. I understand your point, maybe an ablation could illustrate this point and remove the possible interrogation for the reader.
> > > Moreover, I was wondering, what happened if one uses PRDP at inference time? i.e. a NN is trained using PRDP to reduce training time, what happens if the inner loop (which is akin to the inference step?) is partially optimized for example?
> > > Is it applicable to some fine-tuning steps? Are the performances better using a PRDP training than without? (because the network has been trained on less complete training, maybe one can hope for an improvement for partially finished inner loop at test-time also?)
> > > 11. Thanks for your answer, maybe you could use another term than "real-world" to designate this dataset , to avoid misunderstanding for the reader, who could look for real-world measurements rather than synthetics data. Real-world data are often incomplete, noisy and imperfect, so it raises new issues during training.
> > >
> > > Based on the answers provided by the author, I think this paper is interesting. I will raise my score to 6 and vote for acceptance *with the added modifications we've discussed about*, i.e. explaining more clearly the role of PRDP during training (q1-3), adding training time comparison*. I think most of my questions would have been answered with a more straightforward description of PRDP.

---

> > > > ### Author Response · Authors · 2024-11-26
> > > >
> > > > Dear reviewer,
> > > > We have uploaded a revised manuscript. As requested, we have added:
> > > > - The exact numbers for neural networks' performance trained with converged physics vs. PRDP: in section G.2.
> > > > - A study on the neural network’s performance and PRDP savings with increasing network size: in section G.3.

---

> > > > > ### Comment · Reviewer_h4zx · 2024-11-28
> > > > >
> > > > > I thank the author for their responses and for the additional experiments during this discussion period.
> > > > > 7.b That is what I meant. Thanks for this additional explanation. This answers my question.
> > > > >
> > > > > With these additional explanations and experiments, I think this paper can be accepted at the conference (6).

---

> ### Author Response · Authors · 2024-11-25
>
> Dear Reviewer,
>
> Thank you for your thoughtful feedback and for raising your score. We have incorporated your recommendations into the revised manuscript to improve clarity and presentation. Specifically:
>
> - **Clarity Enhancements**: We added multiple schematics (new Figures 1, 5, and 9) and included pseudo-code (Listing 1) to clarify the integration of the neural network within the framework.
> - **Nomenclature**: A detailed nomenclature section is now included in Appendix A to improve readability.
>
> Below, we address your specific replies:
>
> 6. **Adding training duration**: We have added wall-clock training times for PRDP versus fully refined physics in Figure 24 and Section G.1, and the relevant savings% in the main text.
> 7. Addressing your points individually:
>
>     **a. Ablation Study on Training Loss as a PRDP Performance Indicator**: We are compiling the data and will include the corresponding plots in the final revised PDF tomorrow.
>
>     **b. Using PRDP at inference**: Since there is no outer optimization during inference (the network is already trained), we assume your question refers to leveraging incomplete convergence (IC) savings during inference. Specifically, if PRDP terminates refinement at $K_{\text{max}}$ during training, can this level of refinement be used for inference?
>
>     - This is an intriguing idea, and we have added it to the outlook section.
>     - In general, if inference conditions match the validation metric computation (e.g., same initial condition distribution and number of unrolled steps), $K_{\text{max}}$​ may suffice without degrading performance.
>     - However, practical inference conditions often differ from training, so reduced refinement could negatively impact generalization. For robust performance, we recommend full refinement during inference.
>     - Note that only the last experiment of our manuscript (section 4.4 on neural-hybrid emulators) is a setting that involves a physics solver during inference. The pure prediction neural emulator for Heat and Burgers and the inverse problem for the Poisson equation do not. Hence, there can not be any IC savings during inference since there is no iterative process during inference.
>
> 11.We have removed the term “real-world” from the introduction to avoid misunderstandings.

---

> ### Author Response · Authors · 2024-11-28
>
> Dear reviewer,
>
> We are glad that you found our answers helpful. Thank you for your vote of acceptance.
>
> We have just uploaded the final revised pdf with the complete set of supplementary results. We briefly summarize them here:
>
> * Network expressiveness and PRDP savings: We conducted an ablation for the experiment on emulating the Heat PDE 1D. For this, we increased the emulator's parameter space by one order of magnitude. This improves the final validation accuracy (due to the increased capabilities of the model) but only marginally affects the PRDP savings. Most importantly, it does not lower the IC savings, they are persistent across the varying network sizes.
>
> * To answer your original question: _"For the IC savings ... what if the neural network size increases/its expressiveness improves? Are the performances better?"_
> Our ablation shows that the accuracy of the outputs improves, while the IC and PR savings remain persistently high. Together they reduce the iteration count by 80% across the three network sizes.
>
> * Running PRDP on the training loss: We repeated Heat 2D, Heat 3D, Burgers and Navier-Stokes with both training loss and validation loss as the PRDP indicator. The results show that training loss can also serve as a criterion for stepping in PRDP. The final accuracy achieved of the emulators is similar to the results with validation metrics. Using training losses yields slightly higher PR savings because the refinement is slower: when the validation metric plateaus the training loss often still continues to go down. This is also the observation that underpinned our initial reply above.
> While training loss as the PRDP indicator worked for our experiments, we recommend caution: We expect that for more complex problems, such as those with multi-modality or spurious minima, the training loss will be less reliable than the validation loss. Moreover, the convergence is smoother if the validation metric is used as a PRDP indicator.
>
> We thank you again for your thoughtful questions.

---

> > ### Comment · Reviewer_h4zx · 2024-12-02
> >
> > I thank again author for their responses and additional elements.
> > I will keep my (raised) score to 6 and vote for acceptance during the upcoming discussions.

---

### Official Review · Reviewer_1fLE · 2024-11-01

**Soundness:** 3
**Presentation:** 2
**Contribution:** 3
**Rating:** 6
**Confidence:** 2

**Summary:**

This paper proposes to use progressively refined differentiable physics, termed as PRDP, to increase the training efficiency while not harnessing the accuracy. The key finding lies in the fact that the full accuracy of the neural network is achievable through insufficiently converged solvers. Several experiments are conducted to validate the effectiveness of PRDP in reducing training time.

**Strengths:**

The topic this paper wants to tackle seems interesting. It seems intuitive that, considering the noiseness of neural network training and approximative nature of deep models, the physics solver does not need to fully converge for the network to achieve maximum possible accuracy. This paper proposes to use an adaptive strategy to progressively refine the physics solver and thus improve the training efficiency. Several experiments are conducted to verify the efficacy of the proposed method.

**Weaknesses:**

- This paper should provide more background information about *differentiable physics* to make readers better understand the core contribution of the proposed method. I am not an expert of this field, and I find this paper a little bit hard to follow, and also unaware of the broader context this paper lies in.
- The experiment settings in this paper are not clearly presented. Considering that this is paper submitted to ICLR, I want to know what is the role of the neural networks in each experiment.

**Questions:**

The experiments report the improved efficiency by adopting progressive refinement and incomplete convergence. Does these strategies influence the accuracy?

---

> ### Author Response · Authors · 2024-11-21
>
> Dear reviewer,
>
> Thank you for your valuable feedback. We greatly appreciate that you share our intuition that physics solvers do not always need full convergence to achieve optimal network accuracy. Below, we address your remarks and questions in detail:
>
> 1. **More Background on Differentiable Physics**: We acknowledge that the introduction could better emphasise the broader problem domain requiring differentiable physics. Our work fits into the category of neural emulators, surrogates, or neural operators, which aim to enable efficient forecasting for PDE-governed problems across various scientific and engineering domains. Solving PDEs is fundamental to fields ranging from quantum mechanics (Schrödinger Equation) to structural engineering, fluid dynamics, weather forecasting, climate research, and astrophysics.
>    While many recent approaches are purely neural, hybrid methods that integrate classical numerical solvers with neural components have demonstrated superior performance. For example, these hybrid models have shown success in small-scale fluid problems (e.g., Kochkov et al., Um et al.) and large-scale systems like weather and climate modelling (e.g., NeuralGCM). Notably, the experimental setup in Section 4.4 is conceptually similar to these prior works.
>    Despite their promise, neural-hybrid models face limited adoption due to the computational cost of executing and differentiating through classical solvers during training. Since the majority of compute time in engineering-relevant PDE solvers is spent resolving nonlinear and linear systems, PRDP directly addresses this bottleneck. As we point out in the outlook, PRDP could catalyse broader adoption of differentiable physics. We recognize that this broader context was underdeveloped in the introduction and will revise it to ensure these connections are clear.
> 2. **The role of the neural network**: We apologise for not adequately highlighting the role of neural networks in the experimental setups described in Section 4\. To clarify, neural networks are utilised in three distinct contexts in our work:
>    1. **Neural emulator learning** (introduced in Section 2.3, and used in Sections 4.2 and 4.3): Here, the neural network is trained to *replace* a numerical time stepper, i.e., the simulation method advancing a state in time.
>    2. **Neural correction learning** (used in Section 4.4): In this context, the network is trained to *correct* or modify predictions from a coarse numerical simulator, forming a neural-hybrid emulator.
>    3. **Poisson inverse problem** (introduced in Section 2.2, and used in Sections 4.1): This involves a parameterized right-hand side (RHS). While the RHS in our example is defined by the first three eigenmodes of the Laplace operator scaled by one parameter each, one could alternatively use a neural network to parameterize the RHS in higher-dimensional settings.
>
> 	We agree that the main text could benefit from clearer explanations of these contexts and improved visual aids, such as simplified versions of the flowcharts from Figures 17-19. We will make these changes in the final revision.
>
> 3. **Does PRDP influence network accuracy**: PRDP does not negatively affect network accuracy, which remains consistent within the variability introduced by random seeds (for network initialization and stochastic minibatching). Instead, PRDP improves training efficiency by reducing computational costs while preserving accuracy. We discuss this in Section 2.3 (see also Fig. 3b) and confirm it experimentally in Fig. 4\.

---

> > ### Comment · Reviewer_1fLE · 2024-11-25
> >
> > I thank the authors for providing detailed background of Differentiable Physics, and for detailing the role of the neural networks. Although I think there is still room for improving the clarity of this paper, after reading your responses and also other reviewers' comments, I think this is an interesting and technically solid paper and I will raise my score.

---

> > > ### Author Response · Authors · 2024-11-25
> > >
> > > Dear Reviewer,
> > >
> > > Thank you for your thoughtful feedback and for raising your score. We are pleased that you found the background on Differentiable Physics and the role of neural networks helpful. Based on your and other reviewers' comments, we have worked to enhance the clarity of the manuscript.
> > > For instance:
> > > - Improved Visuals: We added new schematics (Figures 1, 5, and 9) to illustrate key concepts and processes more clearly.
> > > - Pseudo-code: Listing 1 has been added to explicitly highlight where the neural network is integrated within the training pipeline.
> > >
> > > We hope these updates address your concerns and make the paper more accessible and engaging. Please let us know if there are additional aspects we can refine further.

---

### Official Review · Reviewer_5MWA · 2024-11-01

**Soundness:** 3
**Presentation:** 3
**Contribution:** 3
**Rating:** 6
**Confidence:** 4

**Summary:**

This paper introduces a framework to progressively refine the resolution of physics solvers during neural network training. The authors demonstrate that training a neural network with a physics solver scheduled to increase in iterations $K$ over training can significantly reduce computational costs, especially by using fewer iterations in the early training phases (*progressive refinement* savings). Additionally, they observe that a neural network can be effectively trained even when the physics solver has not fully converged, eliminating the need for an extensive number of solver steps to achieve high accuracy (*incomplete convergence* savings). To automate this process and determine optimal parameters for these refinements, the authors propose an algorithm that monitors validation set metrics, incrementally increasing solver refinement when performance plateaus. The framework is validated across four use cases: a linear inverse solver, linear neural emulator learning, nonlinear neural emulator learning, and a neural-hybrid emulator.

**Strengths:**

* The paper is well-written, with clear explanations of the intuitions and motivations behind the method.
* The algorithm is straightforward and effectively delivers the intended results, as seen in the reduction of validation loss with the progressive refinement of the physics solver, particularly notable in Figure 4 for the 2D Heat and 2D Navier-Stokes cases.
* The savings in training time and computational resources are substantial.
* The appendix is thorough and well-organized, with especially valuable details on iterative linear solvers and detailed derivations for each problem.

**Weaknesses:**

* Overall, the technical contribution of the paper is somewhat limited, with the main novelty being the proposed algorithm for iterative refinements.
* All physics solvers employed rely on iterative linear solvers. It would have been interesting to see if the method also applies with other physics solvers.
* With the exception of the final example (the neural-hybrid approach), the other examples appear to be simplified or illustrative cases without clear, concrete applications.

**Questions:**

* How sensitive are the parameters of PRDP? Did you experiment with many different parameters for each problem before achieving the results, or was it relatively straightforward?
* Do you expect the method to achieve similar savings in training time for domains with complex boundary conditions and irregular meshing?
* Do you think your method could work where the physics solver contains incomplete physics (e.g., the parameters are not perfectly calibrated, and some terms of the equations could be missing)?
* How would the method be affected by noise in the observations?
* How well do you think this could help reduce the training time of neural GCMs [1]?

[1] Kochkov et al. Neural general circulation models for weather and climate. Nature, 2024.

---

> ### Author Response · Authors · 2024-11-21
>
> Dear Reviewer,
>
> Thank you for your thoughtful evaluation of our paper. We are delighted that you found our algorithm straightforward, appreciated the substantial savings, and enjoyed the writing. Below, we address your remarks and questions in detail:
>
> 1. **Limited technical contribution**: A key contribution of our paper lies in demonstrating that PR and IC savings exist in a differentiable physics setting. While we agree that the proposed approach—progressively increasing the number of solver iterations—is simple, we view this simplicity as a strength. It underscores that these savings are accessible without requiring highly complex modifications to existing workflows.
> 2. **Applicability to other physics solvers**: This is an excellent point. While we focus on linear and simplified nonlinear systems (e.g., Burgers' and Navier-Stokes equations, solved with a one-step Picard approximation akin to an Oseen problem as per Turek 1999), the extension to fully nonlinear systems is indeed a natural next step. Fully resolving nonlinear systems (e.g., through Newton-Raphson methods) introduces a non-quadratic loss landscape, which may require specific adaptations to PRDP. However, PRDP could potentially be applied to schedule the number of nonlinear solver iterations (e.g., Picard or Newton steps), yielding similar savings. For context, incomplete resolution of nonlinear residuals is a common practice in numerical methods like the PISO algorithm for Navier-Stokes.
>    Beyond nonlinear systems, we believe our work already covers a broad range of linear cases, including symmetric positive definite matrices, asymmetric matrices, parameterized matrices, and saddle-point problems. Are there specific solver types or problem classes you had in mind that we did not address?
> 3. **Applicability of the considered examples**: We deliberately chose simpler examples to illustrate the key mechanisms behind PR and IC savings. As noted by other reviewers, differentiable physics is a technically challenging domain, and simpler cases provide a clear view of these mechanisms. We aim to clarify in the paper that scaling up to higher resolutions (e.g., transitioning from 2D to 3D, or to larger-scale systems) builds directly on these foundational insights.
> 4. **Sensitivity of the PRDP parameters:** For most cases (e.g., 1D/2D Heat and Navier-Stokes), selecting hyperparameters was straightforward, and we experimented with a limited range of values around the defaults specified in the pseudo code (ref. algorithm 1). However, the Burgers case required more extensive tuning. We acknowledge that parameter sensitivity can vary depending on the problem, and we will ensure this aspect is discussed in the final version.
> 5. **PRDP on domains with complex BCs and irregular meshing**: We expect PRDP to generalise to such settings. The IC savings are rooted in the phenomena described in Section 2.3 and should persist in more complex configurations. Similarly, PRDP schedules solver refinement efficiently, approaching the necessary $K\_{\\text{max}}$ for convergence.
>    However, the applicability of PRDP depends on the success of the underlying differentiable physics process. If a linear system cannot converge due to issues like poorly conditioned matrices (e.g., from stretched meshes or difficult boundary conditions), differentiable physics (and thus PRDP) would also struggle. Conversely, for reasonable meshing and boundary handling, the behaviour of the system matrix should align with our simpler experiments, avoiding worst-case scenarios where refinement is forced to full resolution.
> 6. **Could PRDP work under incomplete physics:** PRDP’s performance likely depends on the numerical characteristics of the incomplete physics. If the incomplete physics minimally impact the system matrix spectrum and allow for primal solutions, PRDP should remain effective. Initial stabilisation might require higher $K\_0$ values, slightly reducing PR savings, but IC savings (the dominant contributor) could compensate or even improve in such cases.
> 7. **How does noise in observations affect PRDP:** Noise introduces similar challenges as incomplete physics. Higher initial stabilisation costs might marginally impact PR savings, but IC savings could mitigate this. Analogously to geometric vs. algebraic multigrid methods, PRDP operates effectively at the numerical level, and modest noise levels should not undermine its utility.

---

> > ### Author Response · Authors · 2024-11-21
> >
> > 8. **How well would PRDP reduce training time of neural GCM:** Thank you for raising this interesting question. PRDP is applicable in scenarios involving iterative linear solvers, which are often required in implicit time integration schemes or when solving Poisson problems as part of incompressible Navier-Stokes formulations (found in engineering-scale simulations ,e.g., classical aerodynamics). However, for the NeuralGCM described in [https://arxiv.org/pdf/2311.07222](https://arxiv.org/pdf/2311.07222), our preliminary analysis suggests that linear systems are solved spectrally, bypassing iterative solvers. If so, PRDP would not directly benefit this framework. We will expand the paper’s limitations section to acknowledge this case explicitly. That said, PRDP remains highly relevant for other scenarios involving unstructured meshes or complex boundary conditions, where iterative solvers are inevitable. PRDP could provide substantial benefits for training in these scenarios.

---

> > > ### Comment · Reviewer_5MWA · 2024-11-22
> > > **response 2**
> > >
> > > I thank the reviewer for their response. I was curious to know more about GCM because you cited it explicitly in the beginning of Section 4.4. I understand better now the scope of your method. I think it would be beneficial for the clarity of the manusrcript to include for the final version which existing frameworks could directly benefit from your iterative procedure.

---

> > > > ### Author Response · Authors · 2024-11-25
> > > >
> > > > Dear Reviewer,
> > > >
> > > > Thank you for your thoughtful follow-up and for engaging with the details of our work. We have uploaded the revised PDF, which incorporates additional examples and addresses your feedback comprehensively. Below, we respond to your points in detail.
> > > >
> > > > > Thank you for your response. Regarding the second point, I think my question was simpler than that. I was trying to understand whether your method with iterative refinement could also apply to solvers that have an explicit time-stepping, without involving a linear-solve. In such case, the resolution would be the level of spatial and temporal discretization.
> > > >
> > > > We apologize for misunderstanding your earlier question. You are correct: in the case of purely explicit time-stepping (without constraints like incompressibility), no linear solve is required, and therefore PRDP as described for iterative linear solvers is not applicable.
> > > > We have now clarified this limitation explicitly in the Limitations Section of the revised manuscript. However, your suggestion to explore PRDP at the level of spatial and temporal discretization is an exciting avenue for future research, and we have added this perspective to the Outlook Section.
> > > > It’s worth noting that the neural models we employ—fixed-size MLPs and convolutional networks (e.g., feedforward ConvNets and ResNets)—are not resolution-agnostic. Their performance degrades on resolutions other than the training resolution (c.f. [this lecture slide](https://ethz.ch/content/dam/ethz/special-interest/math/applied-mathematics/camlab-dam/documents/AISE2024/AISE24%2011%20Introduction%20to%20Operator%20Learning%20Part%202.pdf) on page16). To apply PRDP over spatiotemporal resolution, resolution-agnostic neural operators like the Convolutional Neural Operator (CNO) [1] could be more suitable. While this direction remains very interesting, it also involves higher engineering effort, as it requires managing reference data across multiple resolutions. For this work, we focused on scheduling linear solver refinements, where we observed network performance improvements that scale predictably with refinement levels, enabling PRDP.
> > > >
> > > > [1] Raonic, B., Molinaro, R., De Ryck, T., Rohner, T., Bartolucci, F., Alaifari, R., Mishra, S. and de Bézenac, E., 2024. Convolutional neural operators for robust and accurate learning of PDEs. Advances in Neural Information Processing Systems, 36.
> > > >
> > > > > I thank the reviewer for their response. I was curious to know more about GCM because you cited it explicitly in the beginning of Section 4.4. I understand better now the scope of your method. I think it would be beneficial for the clarity of the manusrcript to include for the final version which existing frameworks could directly benefit from your iterative procedure.
> > > >
> > > > Thank you for highlighting this. Our citation of NeuralGCM in Section 4.4 was intended as a broader motivation for neural-hybrid emulators (due to being a recent success story) but could indeed imply that PRDP is directly applicable to it. We have removed the NeuralGCM citation from this section to avoid confusion.
> > > >
> > > > Instead, we now cite Kochkov et al. (2021) and Um et al. (2020), both of which involve solving Navier-Stokes equations with iterative pressure Poisson solvers—cases where PRDP can be applied directly. Both Kochkov et al. (2021) and Um et al. (2020) are examples of training neural models end-to-end with differentiable simulators. The software package jax-cfd introduced by Kochkov et al. (2021) has found usage for other research on neural network-based turbulence models, for example in Shankar et al. (2023) [2]. As long as its Finite-Volume backend (and not its spectral backend) is used, there will always be a linear solve due to the pressure-Poisson problem. Hence, PRDP is applicable.
> > > >
> > > > [2] Shankar, V., Maulik, R. and Viswanathan, V., 2023. Differentiable turbulence ii. arXiv preprint arXiv:2307.13533.

---

> > > > > ### Comment · Reviewer_5MWA · 2024-12-02
> > > > > **Response**
> > > > >
> > > > > Dear Authors,
> > > > >
> > > > > Thank you for your detailed responses and for the effort you have put into revising the manuscript. I appreciate the clarifications provided, particularly regarding the applicability of your method to explicit time-stepping solvers and the thoughtful adjustments made to address my feedback.
> > > > >
> > > > > I am satisfied with your responses and the revised manuscript. At this stage, I will maintain my score and vote for acceptance. I may reconsider (i.e. increase) my score during the reviewer discussion phase.

---

> > ### Comment · Reviewer_5MWA · 2024-11-22
> > **response**
> >
> > Thank you for your response.
> > Regarding the second point, I think my question was simpler than that. I was trying to understand whether your method with iterative refinement could also apply to solvers that have an explicit time-stepping, without involving a linear-solve. In such case, the resolution would be the level of spatial and temporal discretization.

---

### Official Review · Reviewer_Skvr · 2024-11-04

**Soundness:** 3
**Presentation:** 4
**Contribution:** 3
**Rating:** 8
**Confidence:** 4

**Summary:**

This article presents a method to reduce the computational costs of end-to-end training with a linear PDE solver in the pipeline. In particular, the scope of this article is about linear solves that are big enough to require an iterative solver. It posits that the level of accuracy, that is needed of the forward model evaluation and its gradient, increases during training. The PRDP borrows inspiration from bi-level optimization schemes: the method starts with an inaccurate linear solver (where the number of iterations is stopped too early) and progressively increases the accuracy (or number of iterations in the solver) as the training starts to plateau. The authors provide an algorithm to schedule the number of iteration in function of the validation loss. The authors show computational savings from two major mechanisms: the progressive refinements (the early gradient updates in the training cost less iterations of the solver) and incomplete convergence (where the iterative solver reaches the desired accuracy without needed to train until the number of iterations is sufficient). The computational savings are up to 86% (case of the 2D heat equation if I understand correctly 72% (IC) + 14% (PR) = 86%, but the text only reports 81%), for more complicated examples, the savings go down to 59%.

**Strengths:**

This article is clearly written and introduces a promising solution to computational costs of training campaigns that include a linear PDE solver, end to end.
The main result, that progressively increasing the accuracy of the solver in end-to-end training converges to similar performance as a fully converged solver while substantially reducing computing costs, is original.
The benefit from incomplete convergence is very interesting. Although unexpected, the computational experiments show that they originate most of the computational savings.
It is interesting to see that the computational saving are larger for the 2D heat equations than for the 1D heat equation.
It is an interesting finding that unrolled differentiation which inefficiently accurately differentiate a sequence of iterative approximation performs similarly to the smarter implicit differentiation.

**Weaknesses:**

The paper suffer weak baselines. The main use case of the method is for iterative solvers of large linear models, however, the authors use 1D and 2D examples which would likely be solved efficiently by a direct solver. The authors should at least provide one 3D example with the simple heat equation. The heat equation itself is also a weak baseline, more complex linear models such as Helmholtz equation could be considered.

It is worrying that the benefit of the method diminish as the problem get harder as in the nonlinear neural emulator learning. Some discussion is needed about the potential harmful mechanisms that limited the computational savings in that case.

**Questions:**

Please add a computational experiment that scales where an iterative solver is necessary (e.g. 3D heat equation).
Please consider harder baselines for the linear emulator learning, such as Helmholtz equation.
Please add a discussion about potential mechanisms specific to nonlinear neural emulator learning that would explain the reduced computational savings.

Minor:
If needed, please correct the reported saving of the 2D heat equation to 72%+14%=86% as well as the maximum saving in the conclusion.

---

> ### Author Response · Authors · 2024-11-21
>
> Dear Reviewer,
>
> Thank you for your constructive feedback on our manuscript. We are pleased that you found PRDP to be an original contribution. Below we address your remarks and questions in detail:
>
> 1. **The choice of model/experimental complexity and baseline:** We chose the set of experiments specifically to illustrate the key aspects of PRDP (progressive refinement and incomplete convergence savings) as well as showcase how PRDP is applicable in different problem scenarios. The experiments thereby show well-behaved symmetric positive definite linear matrices, asymmetric upwinding matrices, parameter-dependent matrices and saddle-point problems. We acknowledge that under the investigated resolutions, a direct solver would generally be preferable. Yet, we think that the savings achievable with PRDP should also exist for higher spatial resolutions (hence larger system matrices). This is because when resolutions of PDE discretizations (on uniform Cartesian grids) grow, sparsity patterns and the form of the spectrum stay almost the same, albeit the condition number grows indicating a slower convergence (-\> requiring more linear solver iterations, but PRDP likely will deliver similar percentage savings). We appreciate your feedback and executed a 3D heat emulation experiment (for results, see point 3).
> 2. **Differences in PRDP savings over increasing experimental difficulty**: Thank you for raising this interesting observation. Training a neural network for the Burgers setup was particularly challenging. When training was performed using very coarse physics (less than 4 solver iterations), the training severely diverged. We started training with a relatively high level of refinement for this case, hence the lower savings. While divergence may reduce PRDP’s benefits, problems of increasing complexity do not necessarily pose inherent issues. For instance, in the correction learning experiment for Navier Stokes, PRDP enabled nearly the same saving as the heat 1D/2D emulator training experiments. Additionally, we have added a 3D heat emulator learning example as highlighted below.
> 3. **Computational Experiment where an iterative solver is necessary**: We have added a three-dimensional heat emulator learning example and can confirm similar savings of 80% (62% IC savings and 18% PR savings). This indicates that similar savings can be expected for larger, more complex problems. We will add the details in the revised pdf.
> 4. **Harder baselines for the linear emulator learning experiment:** The Helmholtz equation is a steady-state equation and in its core formulation, it is an eigenvalue problem. Hence, our heat emulator learning problem does not transfer directly. While we can imagine that PRDP might also be applicable when using iterative eigenvalue solvers (such as the power method or the Lanczos algorithm), this is beyond the scope of the rebuttal. A more trivial extension of the existing experiments would be to perform the Poisson inverse problem instead of the Helmholtz equation with an inhomogeneous parameterized forcing term (similar to the Poisson equation being the inhomogeneous extension to the Laplace equation). Ultimately, the Helmholtz equation (when considered of the form $-\\Delta u \+ k^2 u \= f$) is similar to the Poisson equation (in terms of matrix sparsity pattern and SPD-property). Thus, we think PRDP could potentially also yield benefits in Helmholtz solvers. However, for this rebuttal we have focused on the 3D heat problem, as outlined above.
> 5. **Correction in total savings percentage**: Thank you for pointing out this small typo. We will fix the total savings numbers in the pdf.

---

> > ### Comment · Reviewer_Skvr · 2024-11-21
> >
> > The authors addressed my concerns in a satisfying manner. I appreciate that they added the computational experiment of the 3D heat equation and found similar savings. Provided that these results appear in the revised PDF (which I could not find), I raised my rating from 6 to 8 (accept, good paper).

---

> > > ### Author Response · Authors · 2024-11-25
> > >
> > > Dear Reviewer,
> > >
> > > Thank you for your thoughtful response and for increasing your score. We are delighted that you found our additions, particularly the 3D heat emulator experiment, to be valuable.
> > > We have uploaded the revised PDF, where the 3D heat emulator results are now included in Section 4.2 and Figure 7. Additional details about this experiment are provided in Appendix Sections D.4 and F.2.

---

### Author Response · Authors · 2024-12-01
**General Response**

Dear Reviewers,

We thank you all for your thoughtful feedback and the constructive questions. We are glad that you found the paper thorough and our contributions original and promising.

Below we’d like to summarize the key updates and discussions from the rebuttal.

- We **added results from a 3D case**. We thank reviewer Skvr for the suggestion. Our method **performed equally well** with 81% iterations savings, and **outperformed 1D/2D cases** in compute time savings.
- We highlight the intentional simplicity of our algorithm to ensure accessibility and easy adoption without requiring complex modifications to existing workflows.
- We made several improvements for clarity and presentation. In addition to the main text, we have used rebuttal comments to underscore our contribution - **an improved (cost-saving) training methodology for differentiable physics solvers used in training neural networks**.
- We improved our **introduction to differentiable physics and experimental setups** for readers not familiar with this domain using intuitive visualizations and pseudo-code. Similarly, we have added an **intuitive overview of our proposed algorithm** through a visual and flowchart. These enhancements in the main text supplement the extensive details available in the appendices.
- We included the **wall clock time savings**. Since our method alleviates compute cost by reducing the number of solver iterations, we believe that cumulative number of iterations remains a very good proxy for our method’s performance. The wall clock savings confirm that our algorithm provides substantial speedups, e.g., the training time is reduced by **78%** for the 3D case.

We will continue refining our paper for the camera ready version and will make the experiments’ source code (attached as supplementary material) publicly available upon acceptance. We’d be happy to answer any additional questions that arise.

---

### Meta-Review · Area_Chair_CNDJ · 2024-12-22

**Metareview:**

The paper proposes a method to address the case, when you need to solve end-to-end learning problems with a linear solver in the pipeline. For the case when the linear system is too big, it can be only solved approximately, and the question is to what accuracy we need to solve such kind of systems. The authors provide an algorithm to schedule such kind of tolerances and show improvement in the computational speed. All reviewers agree that this is a good paper.

**Additional Comments On Reviewer Discussion:**

In the rebuttal, results for the 3D problem have been added and included wall-time savings, which is a must for such kind of paper and research.

---

### Decision · Program_Chairs · 2025-01-22

Accept (Poster)